# Cell type-specific delivery by modular envelope design

Daniel Strebinger [1,2,3,4,5], Chris J. Frangieh[1,2,3,4,5,6], Mirco J. Friedrich[1,2,3,4,5], Guilhem Faure[1,2,3,4,5], Rhiannon K. Macrae[1,2,3,4,5] & Feng Zhang [1,2,3,4,5] ✉

The delivery of genetic cargo remains one of the largest obstacles to the successful translation of experimental therapies, in large part due to the absence of targetable delivery vectors. Enveloped delivery modalities use viral envelope proteins, which determine tropism and induce membrane fusion. Here we develop DIRECTED (Delivery to Intended REcipient Cells Through Envelope Design), a modular platform that consists of separate fusion and targeting components. To achieve high modularity and programmable cell type specificity, we develop multiple strategies to recruit or immobilize antibodies on the viral envelope, including a chimeric antibody binding protein and a SNAP-tag enabling the use of antibodies or other proteins as targeting molecules. Moreover, we show that fusogens from multiple viral families are compatible with DIRECTED and that DIRECTED components can target multiple delivery chassis (e.g., lentivirus and MMLV gag) to specific cell types, including primary human T cells in PBMCs and whole blood.

A critical unmet need in the field of gene therapy is the ability to robustly deliver biological cargoes, such as DNA, RNA, protein, or ribonucleoproteins (RNPs), to a broad range of target cell types[1]. To date, adeno-associated viral (AAV) vectors have achieved some tropism, for example, for myocytes[2] and the brain[3], but many tissues cannot be targeted, and bioaccumulation in the liver remains problematic. Alternative delivery approaches using elements from enveloped viruses, such as lentiviral vectors[4], nanoblades[5], eVLPs[6], engineered exosomes[7], or endogenous retroviral proteins[8], offer platforms for engineering tropism via pseudotyping, a process in which a viral fusion protein (or fusogen) is presented on the surface of the vector particle[9,10]. The fusogen fulfills two major roles: First, it allows the viral membrane to fuse with the host cell membrane, thereby releasing the cargo into the cytoplasm of the target cell, and second, it determines the tropism of particles, usually by interacting with specific receptors[11,12].

The most commonly used envelope protein for pseudotyping is the G protein of vesicular stomatitis virus (VSV-G). VSV-G has a broad host cell tropism due to its interaction with low-density lipoprotein receptor (LDL-R) and possibly other LDL-R family members, which are widespread on disparate cell types[13–15]. A previous study successfully incorporated an anti-major histocompatibility complex I (MHCI) single-chain variable fragment (scFv) into the N-terminus of VSV-G. Pseudotyping lentivirus with this engineered VSV-G led to particles that preferentially transduce human cells over mouse cells, but this variant reduced the titer, a critical parameter for translational use where high titers are essential[16]. Furthermore, recent work established a VSV-G variant that has lost affinity for LDL-R, which, when co-expressed with MHC-peptide complexes, could be used for the specific identification of antigen-specific T cells[17,18].

Many other viruses use fusogens to achieve specific tropism. Beyond VSV-G, several other viral envelope proteins have been explored for their ability to target specific cell types, such as HIV-1 env for CD4+ T cells[19], Baboon endogenous retrovirus envelope glycoprotein for human T, B, and CD34+ cells[20], or rabies virus glycoprotein for neural cells[21]. Moreover, some of these alternative fusogens have

[1]Howard Hughes Medical Institute, Cambridge, MA 02139, USA. [2]Broad Institute of MIT and Harvard, Cambridge, MA 02142, USA. [3]McGovern Institute for Brain Research, Massachusetts Institute of Technology, Cambridge, MA 02139, USA. [4]Department of Brain and Cognitive Sciences, Massachusetts Institute of Technology, Cambridge, MA 02139, USA. [5]Department of Biological Engineering, Massachusetts Institute of Technology, Cambridge, MA 02139, USA. [6]Department of Electrical Engineering and Computer Science, Massachusetts Institute of Technology, Cambridge, MA 02139, USA. ✉e-mail: zhang@broadinstitute.org

also been engineered. For example, an antibody binding domain of protein A has been integrated into the sindbis virus envelope, thereby enabling antibody-mediated retargeting[22–24]. Although promising, these chimeric proteins need to be individually engineered for optimal performance[25,26]. Viruses in the *Paramyxoviridae* family, such as measles virus or Nipah virus, naturally separate cell entry and cell targeting into two proteins: the fusion protein (F) is responsible for membrane fusion, and the second protein (either H for measles virus or G for Nipah virus) is responsible for receptor binding[27]. Analogous to the engineering of VSV-G, a targeting molecule was incorporated into the G protein of Nipah virus, which enabled the targeting of specific cell types[28]. However, this strategy has several drawbacks. The use of the F protein in pseudotyping typically results in low titers during production, and the H (or G) protein must be co-engineered to prevent disruption of the interaction with F[29]. Together, these studies highlight the potential of separating the cell entry and cell targeting functions of fusogens.

Here, we describe DIRECTED (Delivery to Intended REcipient Cells Through Envelope Design), a modular platform for achieving programmable cell targeting that separates the fusion and targeting functions of fusogens and can be used with various packaging chassis. DIRECTED encompasses both natural and engineered cell fusion components as well as multiple strategies for cellular targeting. We show that these components can be used with lentiviral particles, which offer the capacity to deliver integrating genetic information, eVLPs, which can deliver Cas9-sgRNA RNPs, and CreVLPs, which deliver Cre recombinase protein. The modular nature of DIRECTED enables precise cellular targeting across diverse contexts and substantially expands the landscape of targetable cell types.

## Results

### Development of DIRECTED

During the assembly of viruses and virus-like particles (VLPs) from cells, membrane proteins can be incorporated into the viral envelope. Production of lentiviral vectors is usually achieved by transient transfection of producer cells with plasmids encoding an envelope protein, such as VSV-G, the cargo of interest, and a helper plasmid encoding lentiviral genes. The resulting viral vectors can then be applied to recipient cells, where the cell targeting and entry are mediated by envelope-receptor interactions and endosomal processes, respectively, resulting in successful transgene delivery to cells expressing the cognate receptor (Fig. 1a). However, for some fusogens, including VSV-G, membrane fusion is induced at low pH and is independent of the engagement of a cellular receptor[30]. Recent work showed that a mutant version of VSV-G can be used in conjunction with additional ligands, such as single-chain variable fragments or peptide MHC single-chain trimers, to retarget lentiviral vectors[17,31]. Thus, we hypothesized that we could expand the tropism of VSV-G by co-expressing a targeting molecule on virions. To generate high-specificity interactions, we focused on using antibodies as an additional targeting moiety. To create a modular system for antibody recruitment, we engineered protein AG (pAG), which binds antibodies, by inserting the coding sequence of pAG between the secretion signal and the transmembrane domain of VSV-G so that it would be secreted and presented on virions. We confirmed this pAG variant was properly localized and retained antibody-binding capacity (Supplementary Fig. 1a).

We initially produced lentiviral particles with an EF1α-driven H2B-mCherry cargo to determine if co-transfection of the pAG construct and a VSV-G construct during production generates functional particles. We transfected different weight ratios of VSV-G to pAG plasmids during lentiviral production, while keeping the amounts of the other components constant. We then evaluated the transduction efficiency of the resulting virions in the presence or absence of an HLA-A2-targeting antibody on HEK293FT cells, which express high levels of HLA-A2 (Supplementary Fig. 1b). HLA-A2 is a membrane protein that

belongs to the human leukocyte antigen (HLA) family. Members of this family form heterodimers with the invariant β−2-microglobulin (B2M) protein to form MHC class I molecules. These molecules are present on the surface of all nucleated cells in vertebrates. If B2M is lost, MHC class I molecules cannot form, and, as a result, they will not be present on the surface of cells. Analysis of the cells by flow cytometry four days after transduction revealed that the addition of the αHLA-A2 antibody resulted in a higher fraction of H2B-mCherry positive cells between the 95:5 and 25:75 ratio of VSV-G:pAG (Fig. 1b, Supplementary Fig. 1c), whereas no increase was observed with B2M knockout HEK293FT cells (Supplementary Fig. 1d). Production of lentiviral particles pseudotyped with wild-type VSV-G leads to production of high lentiviral titers, whereas previously reported modifications to VSV-G have reduced titer[16,32]. To determine if lowering the amount of VSV-G affected production, we quantified the copies of the viral genome in the supernatant after benzonase treatment by RT-qPCR (Supplementary Fig. 1e), finding no significant differences in titers. We then calculated the increase of transduced cells by subtracting the percentage of mCherry-positive cells in the absence of an antibody from the percentage measured in the presence of the αHLA-A2 antibody (Supplementary Fig. 1f). Furthermore, normalizing the transduction efficiency to the viral titers revealed the optimal ratio for antibody-mediated transduction to be between 75:25 and 50:50 of VSV-G:pAG (Supplementary Fig. 1g), which we used for all further experiments. Thus, tropism can be expanded through the addition of a second targeting moiety.

We next sought to block the intrinsic VSV-G tropism through the addition of a soluble competitor molecule[15] (Supplementary Fig. 1h, i) or mutation of the receptor binding domain (Fig. 1c). Based on previous work, we mutated key residues (H8, K47, Y209, and R354) shown to be responsible for the interaction of VSV-G with LDL-R[15]. Evaluation of the resulting physical and functional titer of the single mutants showed a ~50% decrease in production efficiency, and mutation of K47 and R354 resulted in the strongest reduction of infectivity, consistent with previous reports (Fig. 1c). To further reduce the residual infectivity, we combined K47 or R354 mutations with other mutations that resulted in reduced infectivity. We found two double mutants, K47Q/R354Q and H8A/R354Q, showed highly attenuated infectivity while maintaining high production levels. We combined the VSV-G K47Q/R354Q double mutant (from here on VSV-Gdm) with pAG and an αHLA-A2 antibody and tested for infection of HEK293FT cells. This strategy resulted in high transduction of wild-type cells in the presence of the antibody, and minimal transduction without the antibody or on B2M knockout cells (Fig. 1d). These results suggested that VSV-Gdm confers efficient cell entry but abolishes inherent tropism, providing the basis for DIRECTED (Delivery to Intended REcipient Cells Through Envelope Design), a modular platform for cell type-specific delivery wherein fusion and targeting functions are separated.

### Antibody-mediated specific delivery to target cells

We next sought to optimize the antibody amount on the outcome of successful delivery by varying the ratio of antibody to viral particles. Using Jurkat E6 T cells, on which we confirmed surface expression of canonical T cell-specific surface markers (CD3 and CD5; Supplementary Fig. 1j), we titrated the amounts of lentiviral particles decorated with VSV-Gdm and pAG delivering an EF1α-H2B-mCherry transgene versus the amount of antibody (Fig. 2a). In the absence of antibody, no cells were successfully transduced, whereas even low amounts of antibody (0.1 μg/25,000 cells) resulted in effective delivery of the H2B-mCherry transgene and increased up to 0.5 μg/25,000 cells. Overall, DIRECTED was robust over a broad range of antibody amounts (between 0.5 μg/25,000 cells and 2 μg/25,000 cells). To evaluate the specificity of DIRECTED, we established Jurkat E6 and K562 co-cultures at varying ratios, between 1:99 and 99:1, and infected the cells with DIRECTED lentiviral particles in the presence of a Jurkat-targeting antibody (αCD3), in the presence of a K562-targeting antibody (αHLA-

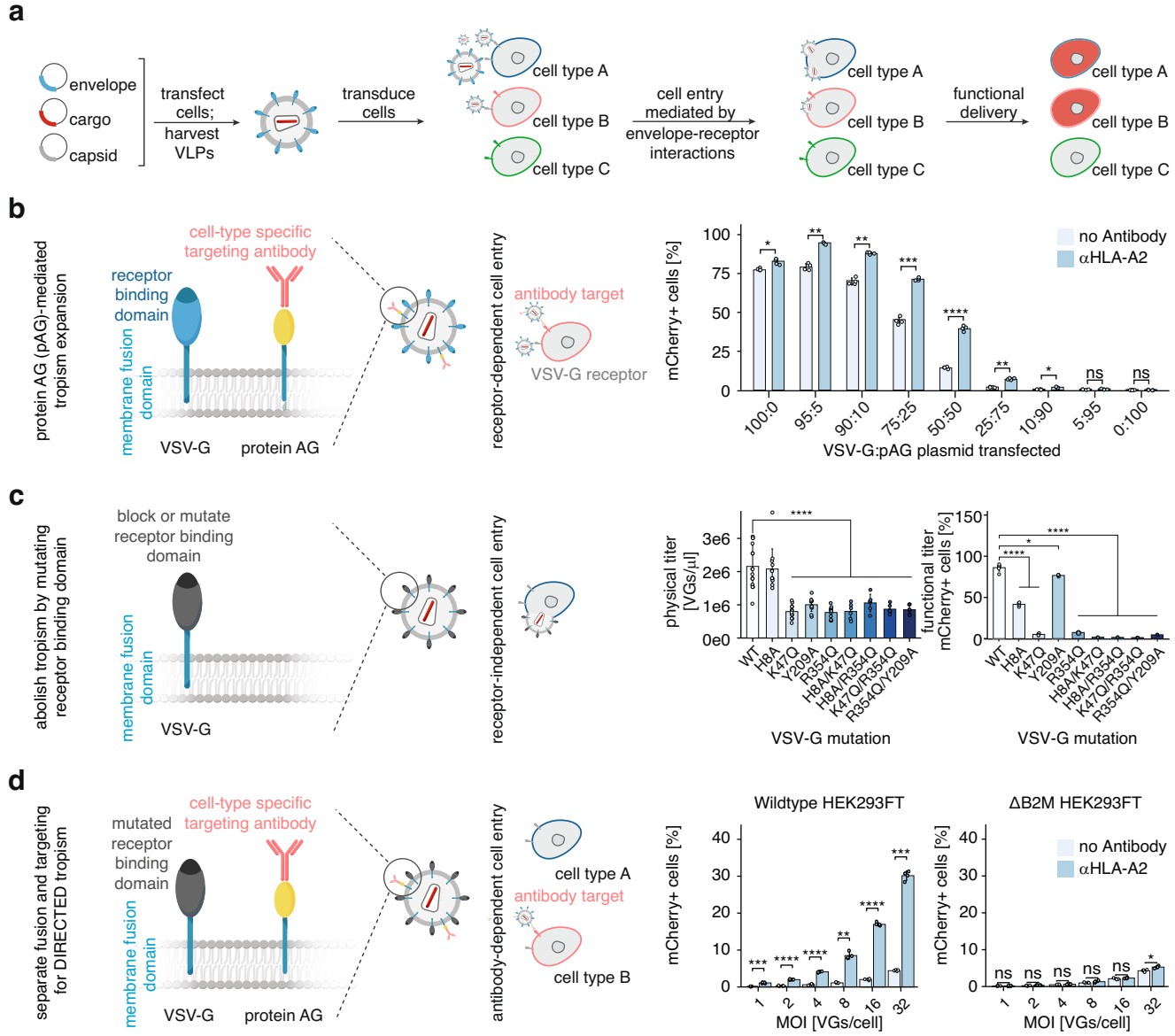

**Fig. 1 | Development of DIRECTED. a** Schematic of lentiviral vector production and lentivector-mediated delivery to target cells. **b** Schematic to expand tropism by co-expressing VSV-G and a variant of protein AG (pAG) that can recruit an antibody on virions (left). Transduction efficiency of HEK293FT cells (as a percentage of mCherry+ cells) of lentiviral particles with various ratios of VSV-G and pAG in the presence or absence of αHLA-A2 (right, $N = 4$ for all conditions, $p$-values: 0.03042, 0.00162, 0.001386, 0.0001269, 7.39E-05, 0.001728, 0.01287, 1, 1). **c** Schematic to reduce VSV-G-mediated tropism by mutating residues that interact with low-density lipoprotein receptors (LDL-R) (left). Quantification of titers for VSV-G mutants (middle; $N = 18$ for WT; 12 for H8A, K47Q, Y209A, R354Q; 6 for H8A&K47Q, H8A&R354Q, K47Q&R354Q, R354Q&Y209A; $p$-values: 1, 2.25E-07, 3.22E-06, 1.92E-07, 1.36E-06, 5.70E-05, 7.31E-07, 6.03E-07). Transduction efficiency of HEK293FT cells (as a percentage of mCherry+ cells) of lentiviral particles with VSV-G point mutations (right; $N = 6$ for WT; 4 for H8A, K47Q, Y209A, R354Q; 2 for H8A&K47Q, H8A&R354Q, K47Q&R354Q, R354Q&Y209A; $p$-values: 2.28E-07, 4.74E-08, 0.016, 5.26E-07, 5.16E-07, 5.22E-07, 5.15E-07, 4.98E-07). **d** Schematic showing separate fusion and targeting components by combining VSV-G K47Q/R354Q (VSV-Gdm) with pAG (left). Transduction efficiency of HEK293FT wild-type cells or B2M knockout cells as a percentage of mCherry+ cells (right; $N = 4$ for all conditions) of lentiviral particles with VSV-Gdm and pAG in the presence or absence of αHLA-A2 at different MOI. ($p$-values: wild-type: 0.0001926, 1.15E-06, 1.54E-06, 0.002304, 1.19E-05, 0.0001458; ΔB2M: 1, 0.258, 0.876, 0.21, 0.822, 0.02598). For panels b, c, and d, a two-sided Welch's t-test with Bonferroni correction was used. [ns, not significant; * $p < 0.05$; ** $p < 0.01$; *** $p < 0.001$; **** $p < 0.0001$; Data are presented as mean ± standard deviation. Source data are provided as a Source Data file].

A2), or in the absence of any antibody (Fig. 2b and Supplementary Fig. 1k, l). We found that even at low target cell ratios, a high proportion of targeted cells were transduced, while off-target infection did not surpass levels observed in the absence of antibody. Our experiments showed that Jurkat cells transduce at a lower efficiency than HEK293FT cells. This may be due to the higher expression of lentiviral restriction factors in myeloid cells[33], and previous studies have shown that the infection rate of suspension cells can be increased using spinoculation[34] and by using transduction enhancers, such as

vectofusin[35] or polybrene[36]. Therefore, we tested these conditions and found that spinoculation combined with 8 μg/ml polybrene significantly increased the transduction of Jurkat cells, while maintaining low non-specific infection (Supplementary Fig. 2a).

## DIRECTED can be used with multiple antibody-immobilization strategies

The modular nature of DIRECTED allows for the use of other targeting strategies in situations where the pAG-antibody targeting approach

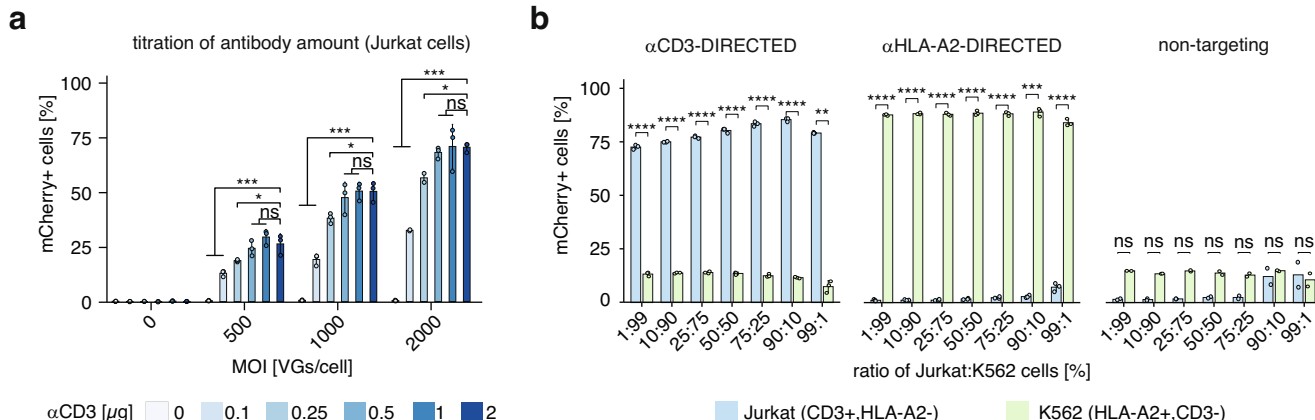

**Fig. 2 | Antibody-dependent specificity of DIRECTED. a** Titration of αCD3 antibody amount for transduction of Jurkat E6 cells ($N = 3$ for all conditions). Data was analyzed by an ANOVA followed by a Dunnett's post-hoc test with a Bonferroni correction. $p$-values: MOI = 0: 0.525515494, 0.996006008, 0.999997333, 0.999917242, 0.237466884; MOI = 500: 5.38E-09, 0.000421347, 0.024090801, 0.841344411, 0.540949719; MOI = 1000: 1.18E-09, 2.30E-07, 0.012478994, 0.856828212, 0.999999895; MOI = 2000: 3.16E-10, 8.30E-06, 0.038954163, 0.96165626, 0.999993745. **b** Transduction efficiency (as a percentage of mCherry+ cells) of Jurkat E6 cells (CD3+) and K562 cells (HLA-A2+) in a co-culture with either

αCD3-DIRECTED ($N = 3$ for each condition), αHLA-A2-DIRECTED ($N = 3$ for each condition), or non-targeting (no antibody, $N = 2$ for each condition) lentiviral particles. A two-sided Welch's t-test with Bonferroni correction was used. $p$-values: aCD3-DIRECTED: 1.95E-05, 8.61E-09, 1.58E-05, 5.52E-05, 4.80E-06, 0.0001883, 6.69E-06; aHLA-A2-DIRECTED: 3.78E-09, 1.32E-06, 2.70E-07, 8.47E-06, 2.81E-07, 2.58E-05, 0.003248; non-targeting: 0.1386, 0.05691, 0.0973, 0.1813, 0.091, 1, 1. [ns, not significant; *$p < 0.05$; **$p < 0.01$; ***$p < 0.001$; ****$p < 0.0001$; Data are presented as mean ± standard deviation. Source data are provided as a Source Data file].

may be inefficient (e.g., in vivo where competitive binding of antibodies found in mouse serum might decrease efficiency). As an alternative to pAG-antibody targeting, we devised an approach for covalent immobilization of targeting molecules at the viral particle surface. We first anchored a single-chain variable fragment (scFv) to the viral particle surface by fusing its coding sequence to the secretion signal and transmembrane domain of VSV-G. We also tested if a SNAP-tag[37], which forms covalent bonds with benzylguaninylated substrates in a process known as click-chemistry, could be used with benzylguaninylated antibodies. To test these strategies head-to-head and evaluate their performance at different receptor expression levels, we engineered a Jurkat cell line expressing a synthetic HA surface receptor (surface-HA), in which we fused an HA-tag to a linker followed by a PDGRFb transmembrane domain (Supplementary Fig. 2b). Using fluorescence-activated cell sorting (FACS), we sorted these cells into four bins (low, medium low, medium high, and high) based on their surface-HA expression level. We then produced DIRECTED lentiviral vectors with each of the antibody mobilization strategies (i.e., scFv, pAG, and SNAP-tag) (Fig. 3a). Western blotting confirmed the presence of the different targeting molecules on viral particles (Supplementary Fig. 2c), and all strategies allowed efficient production of lentiviral vectors as determined by RT-qPCR after benzonase treatment (Supplementary Fig. 2d). To use the SNAP-DIRECTED particles, we first modified commercial antibodies with an NHS-benzylguanine ester, which results in non-specific conjugation of benzylguanine on primary amines, and confirmed the presence of benzylguanine on the antibodies by western blotting (Supplementary Fig. 2e). Next, we incubated the SNAP-tagged lentiviral vectors with a benzylguanine labeled αHA antibody (αHA-BG) for 15 min at room temperature before adding them to cells. To test if excess antibody in the preparation could interfere with the transduction of cells, we subjected an aliquot of the reacted αHA-BG-SNAP-DIRECTED particles to ultracentrifugation through a 20% sucrose cushion, thereby removing excess and unreacted antibody.

We then incubated each of these viral preparations with the four bins of surface-HA expressing cells and wild-type cells (Supplementary Fig. 2f). Using the genetically encoded αHA-scFv decorated particles, we detected high infection rates in all four bins of surface-HA cells, but not in wild-type cells (Fig. 3a, Supplementary Fig. 2g). As expected, pAG and SNAP-DIRECTED lentiviral vectors in the absence of antibody

showed no signal in any of the bins, while we could detect robust signal in the presence of HA-targeting antibodies for both strategies. Interestingly, we found that the transduction efficiency of surface-HA high cells was similar for all tested strategies (Fig. 3, Supplementary Fig. 2h–j). However, we could observe reduced efficiency of pAG-DIRECTED and SNAP-DIRECTED particles on surface-HA low cells in the presence of excess antibody. The infection rate of surface-HA low cells was restored to the same level as observed for the αHA-scFv strategy upon removal of excess antibody for SNAP-DIRECTED particles (Supplementary Fig. 2i). These data suggest that the antibody concentration is a critical variable, especially for the transduction of cells with low surface receptor expression. Moreover, we observed that the transduction efficiency of surface-HA cells plateaued around 30% even at high MOIs (Supplementary Fig. 2k). We speculate that this is due to the synthetic surface-HA receptor, which was generated using a truncated PDGFRb transmembrane domain. In contrast to HEK293 cells[38], PDGFRb is not endogenously expressed in Jurkat cells, which may interfere with the internalization and turnover rate of this receptor on Jurkat cells. As DIRECTED relies on receptors that can efficiently be endocytosed, this could result in inefficient internalization.

## Different antibodies redirect viral tropism efficiently

The ability to immobilize antibodies through various strategies enables a wide range of antibodies, including commercial antibodies, to be used interchangeably to achieve specific cell targeting without the need for protein engineering. To explore the robustness of antibody-mediated DIRECTED delivery, we tested targeting of different synthetic receptors on the same cell type using various antibodies with either pAG-DIRECTED or SNAP-DIRECTED viral particles. We first established cell lines expressing common protein tags as synthetic surface receptors. We then co-incubated these cell lines with pAG-DIRECTED viral particles in the absence of any antibody or in the presence of antibodies targeting the different tags (Fig. 3b). This resulted in successful transduction only for matching surface tag and antibody combinations. Analogously, we interrogated these cell lines with SNAP-DIRECTED viral particles, again observing efficient delivery only for matching surface tag and antibody pairs (Fig. 3c). These experiments also showed that some antibodies perform better using the SNAP strategy than others. We hypothesized that this might result

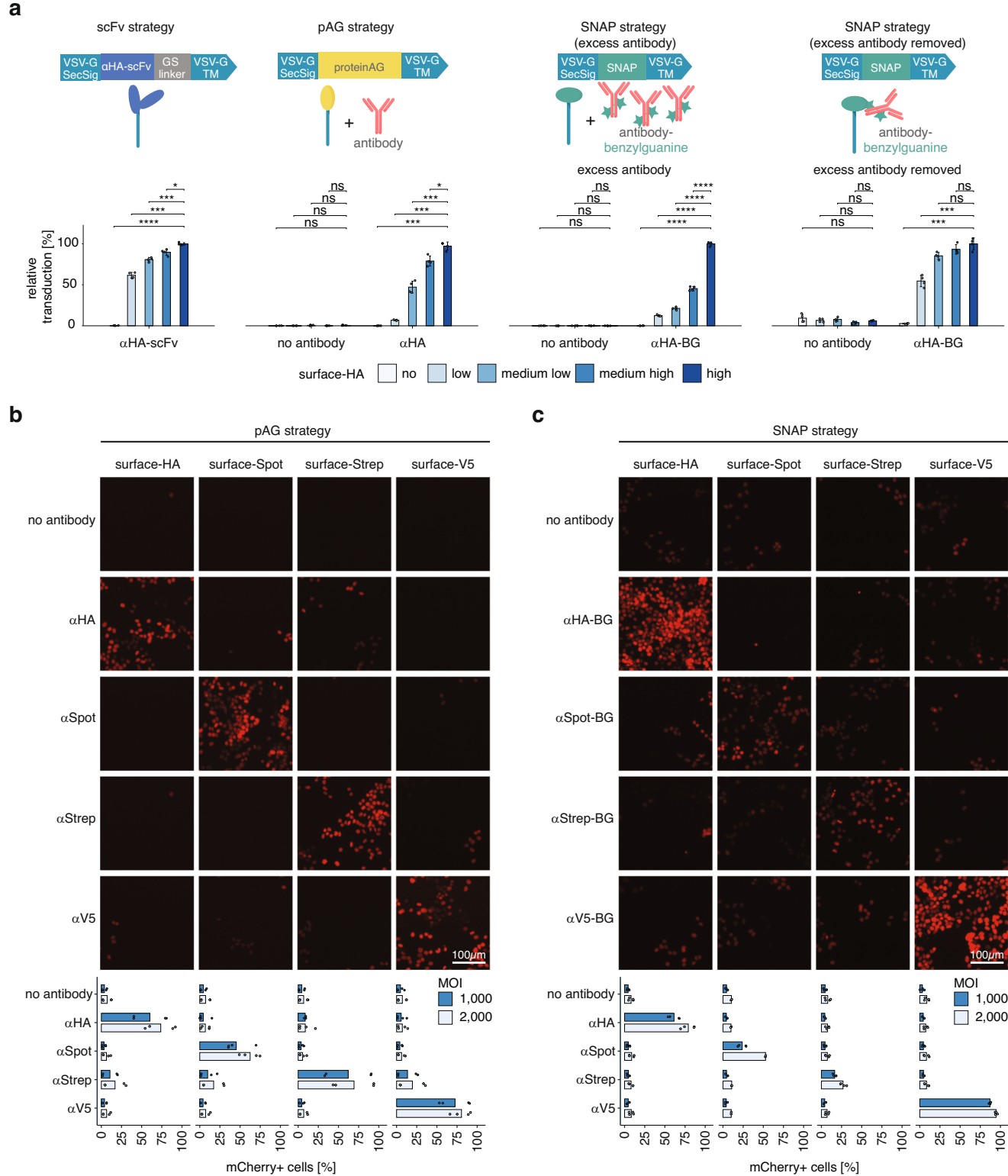

from the NHS ester chemistry used to add benzylguanine (BG) to the antibodies. As the sequence for most commercial antibodies is unknown, it is possible that Lysine residues in the complementary determining regions of the antibodies become modified by BG, thereby inhibiting the binding of these antibodies. To evaluate this possibility, we tested different ratios of BG to antibody for the αSpot antibody and found that this antibody becomes less efficient when modified at higher molar excess (Supplementary Fig. 3a). Finally, we also tested the impact of lentiviral transduction on cell viability and

detected reduced cell counts and an increased percentage of Annexin V positive cells for high doses (MOI = 10000VGs/cell) of wild-type VSV-G pseudotyped lentiviral vectors, but not at the dose ranges used for the experiments presented throughout this study on HEK293FT cells (Supplementary Fig. 3b). Similarly, Jurkat E6 cells showed reduced cell numbers at high doses of wild-type VSV-G (MOI = 500 with spinocu-lation) and a corresponding increase in Annexin V positive cells (Supplementary Fig. 3c). We also noticed an increase in the percentage of Annexin V positive cells with αCD3-targeted SNAP-DIRECTED

**Fig. 3 | Targeting strategies and specificity of DIRECTED. a** (Top) Schematic representation of antibody-tethering strategies and (bottom) relative transduction efficiency of Jurkat+surface-HA cells sorted into four bins of different expression levels (low, medium low, medium high, high) or WT cells (no) using DIRECTED lentiviral particles with the scFv strategy (left), the pAG strategy (middle), or the SNAP strategy (right, with excess antibody or after removal of excess antibody) targeting surface-HA ($N = 4$ for each condition). **b** (Top) Representative microscopy images of HEK293FT cells engineered to express different synthetic surface receptors (surface-HA, surface-Spot, surface-Strep, and surface-V5) after delivery of an H2B-mCherry transgene using pAG-DIRECTED lentiviral particles with the indicated targeting antibodies (no antibody, αHA, αSpot, αStrep, and αV5). (Bottom) Transduction efficiency (as a percentage of mCherry+ cells) for the synthetic cell lines with the indicated targeting antibodies at different MOI ($N = 4$ for each condition). **c** (Top) Representative microscopy images of the same cell lines as in (**b**), using SNAP-DIRECTED lentiviral particles in the presence of an excess of the

indicated BG-labeled targeting antibodies (no antibody, αHA-BG, αSpot-BG, αStrep-BG, and αV5-BG) at different MOI. (Bottom) Transduction efficiency (as a percentage of mCherry+ cells) for the synthetic cell lines with an excess of the indicated targeting antibodies at different MOI ($N = 4$ for each condition). Data are presented as the mean with the error bars indicating the sample standard deviation. Source data are provided as a Source Data file. For panel (**a**) a two-sided Welch's t-test with Bonferroni correction was used. $p$-values sc-Fv strategy 3.52E-06, 0.000116, 0.000242, 0.032; pAG strategy—no antibody 0.368, 0.3, 1, 0.472; pAG strategy—αHA 0.0001224, 0.0001236, 0.00026, 0.016; SNAP strategy (excess antibody)—no antibody 0.524, 1, 1, 1; SNAP strategy (excess antibody)—αHA-BG 1.55E-05, 1.50E-06, 6.08E-08, 3.77E-07; SNAP strategy (excess antibody removed)—no antibody 1, 0.06, 1, 1; SNAP strategy (excess antibody removed)—αHA-BG 0.000412, 0.000384, 0.068, 1. [ns, not significant; *$p < 0.05$; **$p < 0.01$; ***$p < 0.001$; ****$p < 0.0001$; scale bar is 100 μm; SecSig secretion signal, TM transmembrane domain].

particles at an MOI of 50, but not in the absence of an antibody. Evaluating the percentage of Annexin V positive cells as a function of the transduction efficiency (as measured by the Fluorescence Intensity), revealed a trend of increased Annexin V staining with increased transduction rate, suggesting that the uptake of the particles, and the downstream life cycle of the lentiviral vector impacts cell viability of Jurkat cells (Supplementary Fig. 3d). To compare DIRECTED viral particles on the synthetic HA receptor and endogenously expressed receptors, we next tested transduction of Jurkat+surface-HA cells upon targeting of HA, CD3, CD5, or CD46 on these cells using the pAG and SNAP strategy. This showed that although pAG-DIRECTED particles are capable of transducing cells using an αCD3 antibody, there is low efficiency with an αCD5 or αCD46 antibody (Supplementary Fig. 3e). SNAP-DIRECTED particles, on the other hand, showed successful delivery in the presence of αHA-BG, αCD3-BG, αCD5-BG, and αCD46-BG, but not with an αHA antibody that was not modified with BG (Supplementary Fig. 3f). Finally, we tested SNAP-DIRECTED particles on Kasumi-1 cells, a myeloblast cell line that is characterized by high expression of CD117 (KIT), a canonical surface protein usually expressed on hematopoietic stem cells. Targeting of CD117 on these cells resulted in efficient transgene delivery, suggesting that DIRECTED particles can be used on multiple cell types and with different antibodies (Supplementary Fig. 3g, h).

## Fusogens from multiple viral families can be harnessed for DIRECTED

Alternative fusogens from different viral families have been extensively explored for the pseudotyping of lentiviral vectors. Previous work established a receptor-blinded version of the sindbis envelope protein that was further engineered to contain the antibody binding domain of protein A (m168)[23,24], thereby enabling similar modularity as the presented DIRECTED system. We compared these two systems and found that they produce viral particles with similar titers and perform with similar efficiency on HEK293FT+surface-Spot cells (Supplementary Fig. 4a). Although VSV-G pseudotyped vectors are a good choice for in vitro applications, the in vivo function of VSV-G has been reported to be limited due to inactivation by human serum proteins[39,40]. Furthermore, humoral immunity induced by a single round of in vivo injection of VSV-G pseudotyped particles as well as pre-existing immunity against VSV-G could pose a problem for clinical applications[41]. We therefore sought to identify additional pH-dependent fusogens that are compatible with DIRECTED. We performed a sequence-based homology search for VSV-G using psiBLAST. This search revealed multiple sequence-related fusogens from the family of *Rhabdoviridae*, including multiple isolates of vesicular stomatitis virus (Fig. 4a). To evaluate if other members of the family are compatible with DIRECTED, we chose Cocal virus G, which has been used previously to pseudotype lentiviral particles that are more

resistant to human serum inactivation[42]. The predicted structure of Cocal virus G is similar to the predicted structure of a VSV-G monomer (Fig. 4b) and showed major differences only in the flexible region of the Membrane Proximal (MP) domain. Moreover, three of the four residues that are important for the interaction of VSV-G with LDL-R are conserved between VSV-G and Cocal virus G (Supplementary Fig. 4b). Based on the structural comparison, we generated a Cocal virus G receptor-binding mutant predicted to abrogate interaction with the cellular receptor and tested its performance on pAG-DIRECTED lentiviral particles for transduction of HEK293FT cells. In the presence of an αHLA-A2 antibody, we observed high levels of transgene delivery, whereas there was minimal activity in the absence of the antibody (Fig. 4c). To further explore alternative fusogens, we synthesized a library of ~100 fusogens from 11 viral families (Supplementary Data 1). We used this library to produce pseudotyped lentiviral particles (Supplementary Fig. 4c), and then we tested all pseudoviruses on a panel of five cell lines (Supplementary Fig. 4d, e). We found that ~25% of constructs showed transduction of at least one of the tested cell lines, but only a few fusogens showed intrinsic cell type-specific tropism. Next, we tested the library of wild-type fusogens using the pAG targeting strategy on HEK293FT cells. The presence of an αHLA-A2 antibody resulted in an increase in infection for 15 of the tested fusogens, all of which are members of four families (*Filoviridae*, *Orthomyxoviridae*, *Rhabdoviridae*, and *Togaviridae*) and have been previously reported to use a pH-dependent fusion mechanism[43–48] (Fig. 4d). To broaden the types of fusogens compatible with DIRECTED, we focused on one of these families, *Orthomyxoviridae*. Two of the five tested fusogens in this family (Dhori virus GP, DHOV_GP; and a hypothetical membrane protein from Quaranfil quaranjavirus, QRFV_Hyp) showed functional transduction, prompting us to further explore the natural diversity in this family. We performed psiBLAST searches based on DHOV_GP, which revealed QRFV_Hyp and multiple baculoviral envelope proteins, including GP64, which were not in our original library (Supplementary Fig. 4f). Using AlphaFold2[49,50] we predicted the structure of Dhori thogotovirus GP, QRFV_Hyp, and baculovirus GP64 to compare their overall folds. The fusion domain of all three proteins showed high conservation of hydrophobic amino acids in the fusion loop, which allows the protein to insert into target membranes and induce membrane fusion (Fig. 4e). To evaluate the compatibility of GP64 for its use with DIRECTED, we produced lentiviral vectors decorated with unmodified GP64 and pAG and tested them on HEK293FT cells, finding a strong increase in transduction in the presence of antibody (Fig. 4f). Interestingly, GP64, in contrast to VSV-G, has been shown to be suitable for establishing stable producer cell lines as it is not toxic to mammalian cells, highlighting the benefits of using different fusogenic components[51]. Finally, we also compared pAG-DIRECTED particles using GP64, COCVmut, VSV-Gdm, or wild-type VSV-G (without pAG) at the same MOI in the presence or absence

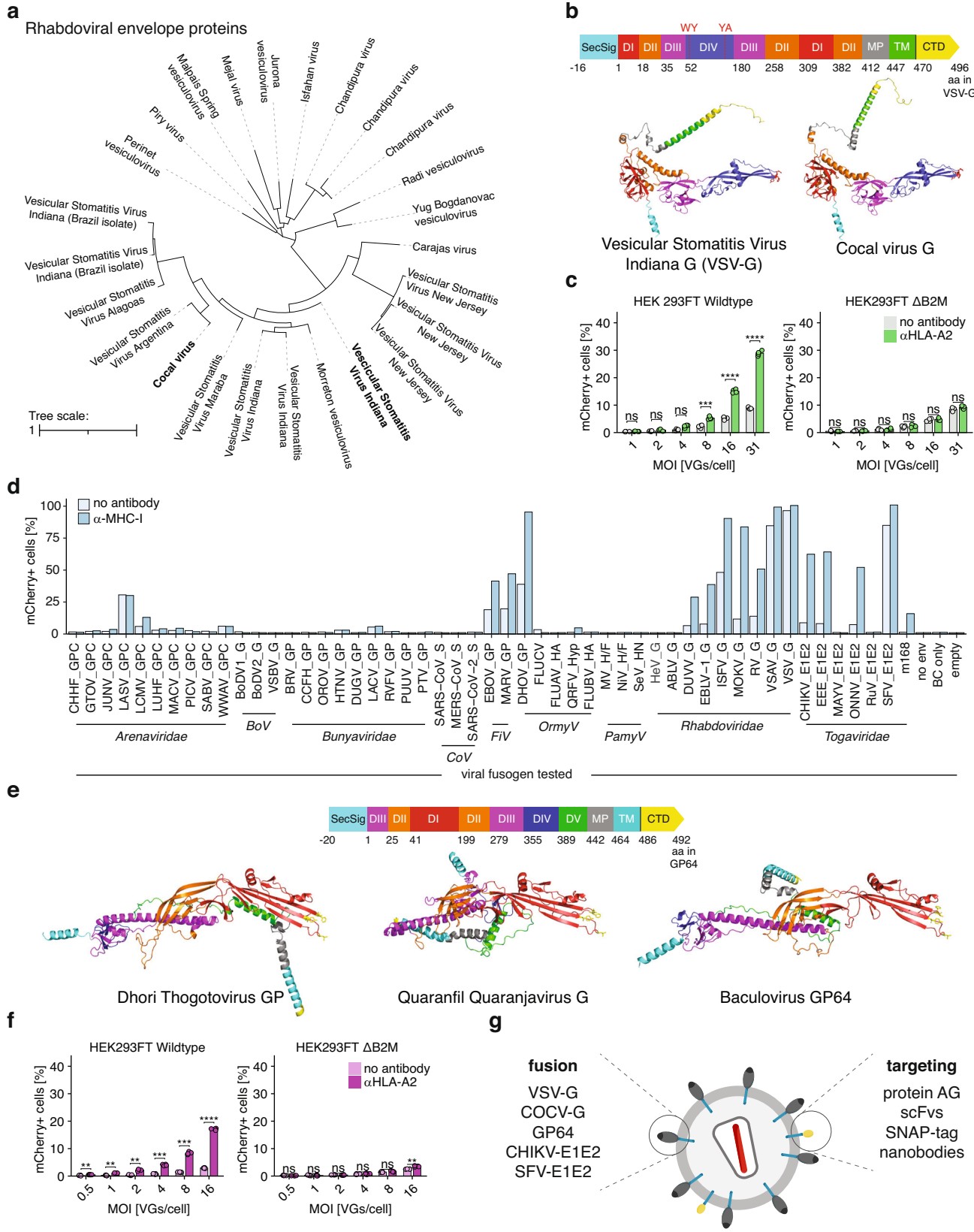

of an αHLA-A2 targeting antibody. This revealed that all envelopes perform at similar efficiency when directed toward HLA-A2 (Supplementary Fig. 4g). These findings demonstrate the flexibility of DIRECTED to work with multiple fusogenic components from different viral families and species in addition to compatibility with multiple targeting components (Fig. 4g).

## DIRECTED works with multiple delivery chassis

Finally, we sought to explore if DIRECTED is compatible with alternative enveloped delivery modalities beyond lentiviral particles. We first evaluated protein delivery using CreVLPs, in which we fused the coding sequence for Cre recombinase to an optimized MMLV gag scaffold[6]. After confirming Cre loading into VLPs (Supplementary

**Fig. 4 | Exploration of the natural diversity of fusogens and their compatibility with DIRECTED. a** Phylogenetic tree of Rhabdoviral envelope proteins using VSV-G as a seed. **b** (Top) Domain structure of VSV-G[52]. (Bottom) AlphaFold2 models of VSV-G and Cocal virus G monomers. Cyan: SecSig, secretion signal; Red: DI, lateral domain; orange: DII, trimerization domain; purple: DIII, pleckstrin homology domain; blue: DIV, fusion domain (hydrophobic residues in red); gray: MP, membrane proximal domain; green: TM, transmembrane domain; yellow: CTD, C-terminal domain. **c** Transduction efficiency (%mCherry+) of WT and ΔB2M HEK293FT cells for pAG-DIRECTED particles using Cocal virus G mutant as the fusogen with an αHLA-A2 antibody at different MOI ($N = 4$ for each condition). **d** Transduction efficiency of HEK293FT cells (as percentage of mCherry+ cells) for a library of pAG-DIRECTED lentiviral particles using the indicated fusogens in the absence of an antibody or with an αHLA-A2 antibody. *BoV−Bornaviridae, CoV− Coronaviridae, FiV−Filoviridae, OrmyV−Orthomyxoviridae, PamyV−Paramyxoviridae*, m168−engineered sindbis envelope, no env−no envelope, BC only− Barcode vector only, empty−no transfer genome transfected. **e** (Top) Domain architecture of GP64[76] and (bottom) AlphaFold2 models for monomers of Dhori thogotovirus GP, Quaranfil quaranjavirus G, and baculovirus GP64 (hydrophobic residues in yellow). Cyan: SecSig; magenta: DIII; orange: DII; red: DI, fusion domain; blue: DIV; green: DV; gray: MP; cyan: TM; yellow: CTD. **f** Transduction efficiency (%mCherry+) of WT and ΔB2M HEK293FT cells for pAG-DIRECTED particles using GP64 as the fusogen with an αHLA-A2 antibody at different MOI ($N = 4$ for each condition). **g** Schematic showing the modular features of DIRECTED particles, highlighting that the fusion and targeting components can be combined in a plug-and-play manner. For panels (**c**), and (**f**), a two-sided Welch's t-test with Bonferroni correction was used. *p*-values in (**c**), HEK293FT WT 1, 0.315, 0.05142, 0.000948, 9.36E-06, 2.35E-05; HEK293FT ΔB2M 1, 1, 1, 1, 0.69; in (**f**), HEK293FT WT 0.004272, 0.001794, 0.00159, 0.00015, 0.000411, 1.99E-06; HEK293FT ΔB2M 0.4314, 1, 1, 1, 0.513, 0.002772. [ns, not significant; *$p < 0.05$; **$p < 0.01$; ***$p < 0.001$; ****$p < 0.0001$] Data are presented as mean ± standard deviation. Source data are provided as a Source Data file.

Fig. 5a) and confirming the functionality of the packaged Cre recombinase in vitro (Supplementary Fig. 5b), we tested if CreVLPs are compatible with DIRECTED by transducing Jurkat E6 cells engineered to harbor a Cre sensitive GFP reporter (Jurkat E6 Cre reporter cells). We found that αCD5-BG SNAP-DIRECTED CreVLPs efficiently induced GFP expression in these cells, whereas we did not detect GFP expression in the absence of antibody (Fig. 5a and Supplementary Fig. 5c). Having established a robust protein delivery system, we then tested the compatibility of DIRECTED with Cas9-RNP delivery by packaging a single guide RNA (sgRNA) targeting B2M into Cas9-RNP VLPs, as previously described[6]. We confirmed high expression of B2M on Jurkat E6 cells by flow cytometry (Supplementary Fig. 5d) and produced particles either with wild-type VSV-G, with VSV-Gdm and pAG, with COCV-mut and pAG, or with GP64 and pAG We then tested these DIRECTED Cas9-RNP VLPs on Jurkat E6 cells in the presence of an αCD5 antibody or in the absence of antibody. To determine the functional knockout efficiency, we used flow cytometry to measure the protein levels of B2M on the cell surface 4 days after delivery. We found that αCD5-DIRECTED particles led to an efficient loss of B2M protein with all tested envelopes, whereas particles without antibody or with non-targeting guides showed minimal knockout efficiency (Fig. 5b, Supplementary Fig. 5e, f). We next wanted to test the ability of pAG or SNAP-DIRECTED particles to target different natural receptors. We started with VSV-Gdm+pAG-DIRECTED particles that target CD3 or CD5 and found that both antibodies were effective at reducing B2M levels compared to the control without antibodies. However, the CD5-directed particles showed lower effectiveness than the CD3-directed particles (Fig. 5c). When we repeated the experiment with VSV-Gdm +SNAP-DIRECTED particles, we could not detect any difference between the two receptors (Fig. 5c). This suggests that the SNAP strategy may be more efficient than the pAG strategy for natural receptors. Finally, we compared the effectiveness of wild-type VSV-G pseudotyped eVLPs with SNAP-αCD3-DIRECTED eVLPs and found that both methods work equally well (Supplementary Fig. 5g, h). These results support our findings on HEK293FT cells that different envelopes that are compatible with DIRECTED perform equally well when used at the same MOI upon targeting highly expressed cell surface receptors (Supplementary Fig. 4g). Together, these results demonstrate that the modular components of DIRECTED are compatible with other delivery chassis beyond lentiviral particles and can be used to achieve high levels of delivery of both protein and RNPs.

## DIRECTED transduces specific cells in a complex environment

To test if DIRECTED lentiviral vectors can be used to target specific primary cells in a complex environment, we purified peripheral blood mononuclear cells (PBMCs), which are composed of multiple different cell types (Fig. 6a). We initially explored whether expansion of the tropism of wild-type VSV-G would allow more efficient T cell targeting, by generating particles decorated with wild-type VSV-G and SNAP, which we then functionalized with an agonistic αCD3-BG antibody. The αCD3 antibody targets a T cell-specific surface protein that is part of the T cell receptor (TCR) and results in T cell activation, which is known to make T cell more susceptible to lentiviral transduction. Six days later, we analyzed the fraction of H2B-mCherry positive cells by flow cytometry, which revealed a significant increase in the fraction of mCherry+ T cells treated with the VSV-G+SNAP +αCD3 over VSV-G, which persisted for at least 14 days (Supplementary Fig. 6a). This shows that the incorporation of activating signals on DIRECTED lentiviral particles can increase the transduction efficiency of primary T cells by coupling cell type targeting and activation.

We then generated DIRECTED lentiviral vectors by functionalizing VSV-Gdm+SNAP particles with ligands targeting cell type-specific surface markers. To target T cells, we again used αCD3 functionalized particles (VSV-Gdm+αCD3) or particles functionalized with a mix of antibodies targeting T cell-specific surface proteins (VSV-Gdm+Tcell), including CD3, CD28 (which provides co-stimulatory signals required for T cell activation), and CD4 (which is expressed on "helper T cells"). Additionally, to target B cells, we functionalized particles with B cell targeting ligands (VSV-Gdm-Bcell), including an αCD19 antibody (targeting a surface marker expressed in all B cell lineages) and MegaCD40L (which simulates the natural activation of B cells). We then transduced PBMCs with these vectors and found that resting human T cells could be efficiently transduced with VSV-Gdm+αCD3 and VSV-Gdm-Tcell vectors, but not with VSV-Gdm+SNAP or VSV-Gdm-Bcell (Fig. 6b and Supplementary Fig. 6b, c), and that transgene expression was stable for at least 14 days (Supplementary Fig. 6d). Moreover, the T cell targeting formulations also outperformed wild-type VSV-G. We also found that VSV-Gdm-Tcell particles resulted in reduced transduction of CD8+ T cells, suggesting the incorporation of an αCD4 antibody resulted in a bias toward CD4+ T cell transduction (Supplementary Fig. 6e).

We then tested targeting of the B cell fraction in PBMCs using B cell targeting ligands (αCD19, and MegaCD40L) on VSV-G+SNAP or VSV-Gdm+SNAP particles (Supplementary Fig. 6f). We found an increase in H2B-mCherry positive B cells in both B cell targeting conditions (Fig. 6c), whereas we could not detect H2B-mCherry positive T cells in the VSV-Gdm-Bcell condition (Supplementary Fig. 6g). The culture conditions we used did not afford the efficient proliferation of B cells, resulting in a low frequency of B cells in population (Supplementary Fig. 6h).

Taken together, these experiments show that DIRECTED allows the incorporation of cell type-specific targeting ligands or combinations thereof, which can increase transduction efficiency when combined with wild-type VSV-G, or completely redirect tropism when using VSV-Gdm.

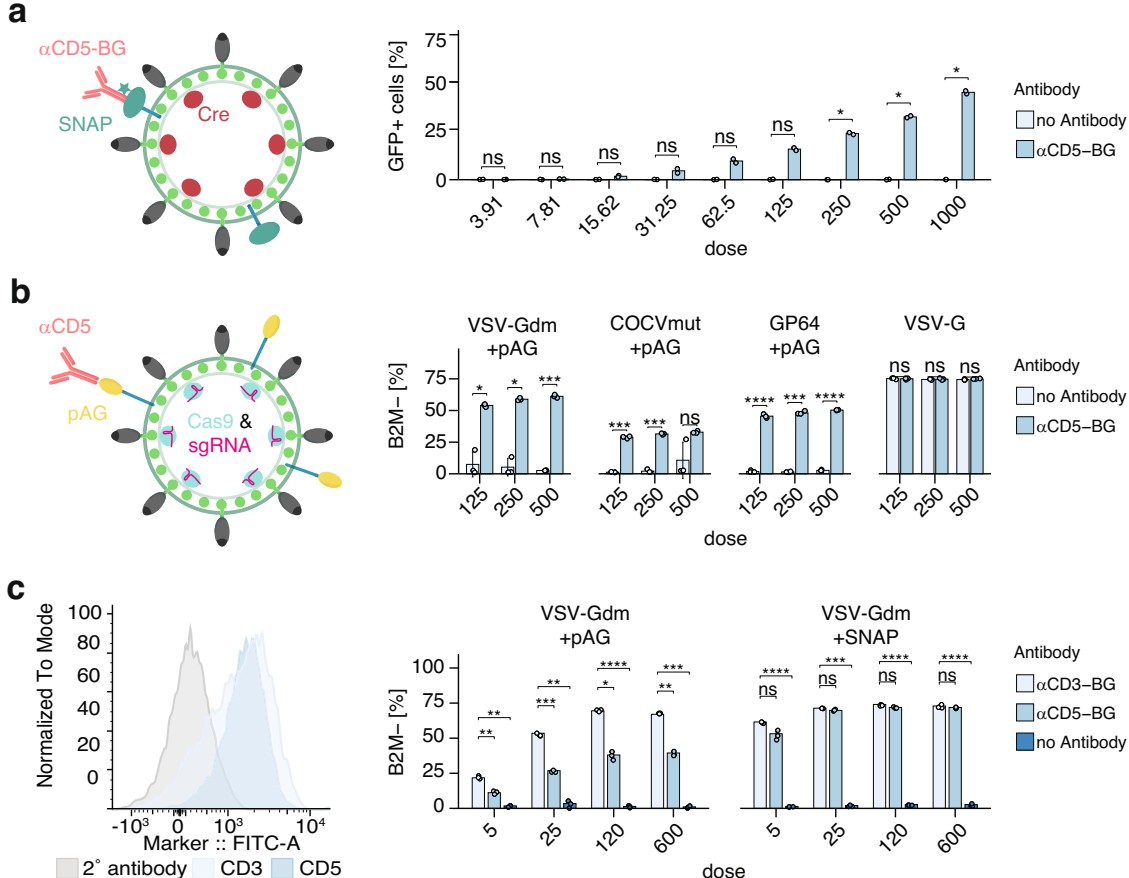

**Fig. 5 | DIRECTED is compatible with modalities that allow protein or RNP delivery. a** (Left) Schematic showing the architecture of αCD5-targeting SNAP-DIRECTED-CreVLPs that package Cre protein. (Right) Efficiency of Cre protein delivery as determined by percentage of GFP+ cells upon application of αCD5-targeting SNAP-DIRECTED-CreVLPs without the removal of excess antibody on Jurkat E6 Cre reporter cells at different doses (*N* = 2 for each condition, *p*-values: 0.328, 0.21375, 0.189, 0.196714286, 0.11088, 0.069075, 0.0435, 0.0435, 0.0435). **b** (Left) Schematic showing the elements of Cas9-RNP delivering pAG-DIRECTED particles. (Right) Loss of B2M protein expression on Jurkat E6 cells upon delivering of Cas9-sgRNA using pAG-DIRECTED particles in the absence of an antibody or with a CD5-targeting antibody for VSV-Gdm+pAG (*p*-values: 0.0408, 0.01419, 0.000408), COCVmut+pAG (*p*-values: 0.000582, 0.000342, 0.345), GP64+pAG (*p*-values: 8.25E-05, 0.000393, 1.15E-05), or WT VSV-G (*p*-values: 1, 1, 0.324) at different

doses (*N* = 3 for each condition). **c** (Left) Histogram of CD3 or CD5 surface expression on Jurkat E6 cells by flow cytometry. (Middle) Loss of B2M surface expression upon treatment of Jurkat E6 cells with VSV-Gdm+pAG-DIRECTED particles in the absence of antibody or upon targeting of CD3 or CD5. [*p*-values: 0.005672, 0.002088, 0.0001464, 0.002368, 0.024, 7.22E-07, 0.008, 0.0001704] (Right) Analysis of the surface level of B2M after using VSV-Gdm+SNAP-DIRECTED particles without removal of excess antibody in the same conditions as before. [*p*-values: 0.552, 5.31E-05, 0.528, 0.0001664, 0.12, 8.56E-05, 1, 1.14E-05] (*N* = 3 for each condition) For panels (**a**), (**b**), and (**c**), a two-sided Welch's t-test with Bonferroni correction was used. [ns, not significant; *\*p* < 0.05; *\*\*p* < 0.01; *\*\*\*p* < 0.001; *\*\*\*\*p* < 0.0001]. Data are presented as the mean with the error bars indicating the sample standard deviation. Source data are provided as a Source Data file.

## DIRECTED enables targeting of T cells in whole blood

We next tested for T cell targeting in whole blood, which adds multiple additional hurdles for gene therapy vectors, such as neutralizing antibodies[41,52], complement components[39], and phagocytotic cells[53] (Fig. 6d). We incubated whole blood with lentiviral vectors pseudotyped either with wild-type VSV-G, VSV-Gdm+SNAP-DIRECTED particles without antibody, or functionalized variants (VSV-Gdm-Tcell: αCD3, αCD28, αCD4; VSV-Gdm-αCD3). After 6 h of co-incubation of the viruses with the blood samples, we removed red blood cells and incubated the purified cells with T cell activator beads, which activate T cells and result in their proliferation and survival. On day 6 post-transduction, we analyzed the cells by flow cytometry and found that both T cell targeting combinations resulted in successful infection of T cells, to a level similar to, or higher than wild-type VSV-G (Fig. 6e, Supplementary Fig. 7a–c). We also confirmed that transgene expression persisted for at least 14 days of culture, indicating successful lentiviral integration (Supplementary Fig. 7a).

Finally, we evaluated SNAP-DIRECTED particles for in vivo applications. To avoid the overstimulation of T cells upon delivering high

amounts of αCD3 antibody, we chose to target CD5, which has recently been used to target lipid nanoparticles (LNPs) to T cells in vivo[54]. We injected Cre reporter animals (Ai9) with αCD5 SNAP-DIRECTED-CreVLPs, SNAP-DIRECTED-CreVLPs without antibody, or VSV-G CreVLPs. Five days post-injection, we analyzed different cell subsets in the spleen for tdTomato expression, which signifies successful delivery (Supplementary Fig. 8a). We found a significant increase in CD3+ cells and a concomitant decrease in CD20+ cells in animals injected with wild-type VSV-G pseudotyped CreVLPs. This change in cell type composition may be indicative of an inflammatory reaction, which we did not observe for DIRECTED CreVLPs. As expected, we found that wild-type VSV-G resulted in the highest overall transduction level and high transduction of all tested cell types, in line with its broad tropism (Supplementary Fig. 8b). Encouragingly, we detected similarly low transduction levels of CD11b+ and CD20+ cells for DIRECTED CreVLPs in the absence of an antibody or upon αCD5 functionalization, suggesting that VSV-Gdm has reduced tropism in vivo. We observed a promising trend for higher transduction of CD3+ cells upon αCD5 targeting, although it was not statistically significant (Supplementary

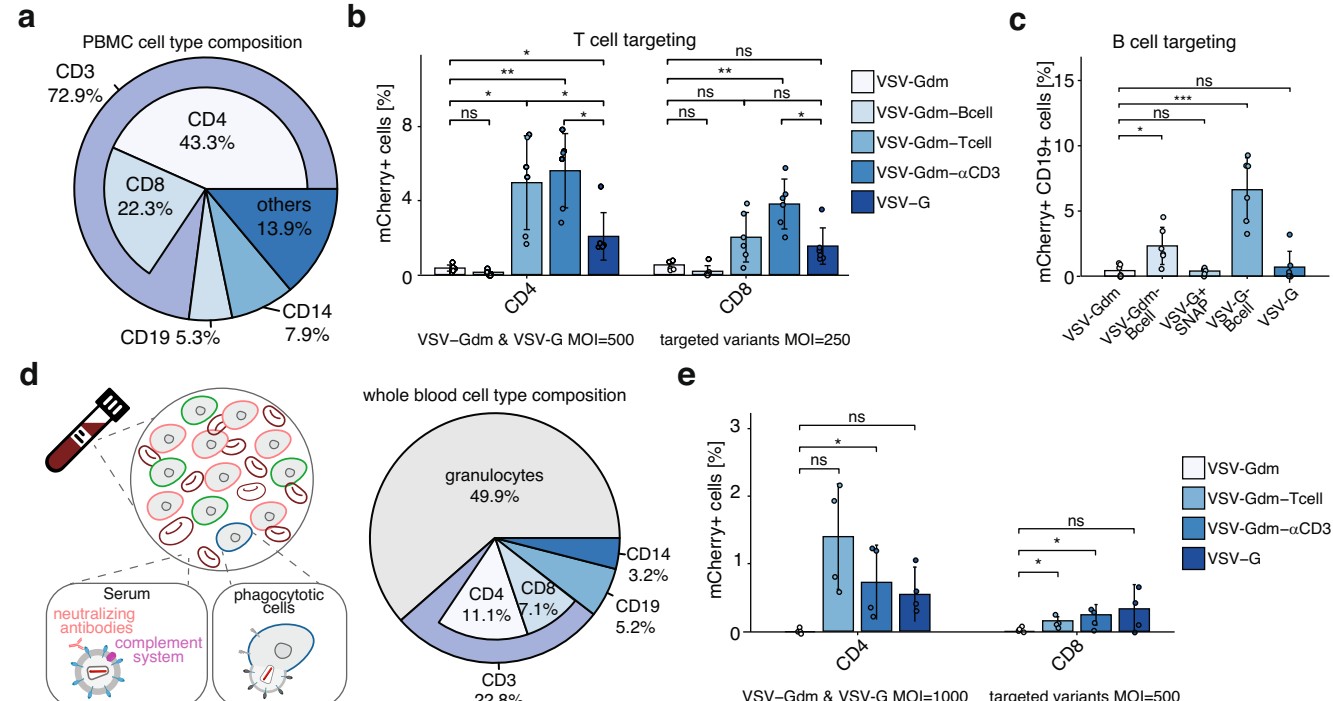

**Fig. 6 | DIRECTED enables specific targeting in complex environments. a** Cell type composition [in percent] of PBMCs as determined by flow cytometry after staining for specific surface markers. **b** Delivery efficiency of H2B-mCherry transgene to primary human T cells in PBMCs from three donors by VSV-G and VSV-Gdm+SNAP lentiviral vectors in the absence of antibody (VSV-Gdm) or functionalized with B cell targeting ligands (αCD19-BG, MegaCD40L-BG; VSV-Gdm-Bcell), T cell targeting ligands (αCD3-BG, αCD28-BG, αCD4-BG; VSV-Gdm-Tcell), or αCD3-BG (VSV-Gdm-αCD3). Shown is the percent of mCherry+ CD4+ (left, *p*-values: 0.083, 0.018666667, 0.004, 0.04, 0.032, 0.032) or mCherry+ CD8+ T cells (right, *p*-values: 0.050285714, 0.050285714, 0.004, 0.050285714, 0.333, 0.032). VSV-G and VSV-Gdm: MOI of 500, other variants: MOI of 250 (*N* = 2 infections per donor for 3 donors) **c** Delivery efficiency of H2B-mCherry transgene to primary human B cells in PBMCs from three donors by WT VSV-G, WT VSV-G+SNAP, VSV-Gdm+SNAP (VSV-Gdm), VSV-G+ αCD19-BG, MegaCD40L-BG (VSV-G-Bcell), and VSV-Gdm+ αCD19-BG, MegaCD40L-BG (VSV-Gdm-Bcell). VSV-G, VSV-G+SNAP and VSV-Gdm: MOI of 3500; VSV-Gdm-Bcell, and VSV-G-Bcell: MOI of 1750. (*N* = 2 independent infections

per donor for 3 donors, *p*-values: 0.021, 0.883, 0.001, 0.627) **d** (left) Schematic highlighting defense mechanisms in whole blood. (right) Cell type composition of whole blood after removal of red blood cells. **e** Delivery efficiency of H2B-mCherry transgene to primary human T cells in whole blood from two donors by VSV-G and VSV-Gdm+SNAP lentiviral vectors in the absence of antibody (VSV-Gdm) or functionalized with T cell targeting ligands (VSV-Gdm-Tcell), or αCD3-BG (VSV-Gdm-αCD3). Shown is the percent of mCherry+ CD4+ T cells (left, *p*-values: 0.079, 0.037, 0.075) or mCherry+ CD8+ T cells (right, *p*-values: 0.014, 0.048, 0.161). VSV-G and VSV-Gdm: MOI of 1000; VSV-Gdm-Bcell, VSV-Gdm-Tcell, and VSV-Gdm-αCD3: MOI of 500. MOIs were calculated estimating 5000 leukocytes per μl of whole blood. (*N* = 2 infections per donor for 2 donors) Data are presented as mean ± standard deviation. Source data are provided as a Source Data file. For panel (**b**), a paired, two-sided Welch's t-test with BH correction was used, and for panels (**c**) and (**e**), a two-sided Welch's t-test was used. [ns, not significant; *p < 0.05; **p < 0.01; ***p < 0.001; ****p < 0.0001].

---

Fig. 8b, c). Interestingly, we did notice splenomegaly in animals injected with any of the CreVLP formulations and could identify an increase in F4/80+ cells in the spleens of these animals (Supplementary Fig. 8d), many of which were tdTomato+ (Supplementary Fig. 8e). These data suggest that DIRECTED enables transgene delivery to target cells in vivo with limited efficiency. The limitations are likely due to macrophage uptake of the injected DIRECTED particles, a key challenge with delivery by many other gene therapy vectors, including other viral vectors and LNPs[55].

## Discussion

Achieving the potential of gene therapy will require a comprehensive array of delivery modalities that can be tailored to specific cargoes and target cells. Here we demonstrate that targeting various delivery modalities, such as lentiviral vectors, CreVLPs, and eVLPS, can be reprogrammed using a suite of modular components. We capitalized on the ability to separate the cell entry mechanisms of these delivery vehicles into independent fusion and targeting functions to generate the DIRECTED platform, which combines interchangeable fusogenic and targeting components.

A key advantage of DIRECTED is the compatibility with commercial antibodies (via both pAG-mediated antibody recruitment or SNAP-

mediated covalent linkage), which substantially expands the range of targetable cell types, including cell types that have been refractory to transduction, such as B cells, which do not express LDL-R. The pAG-mediated strategy, which showed highly efficient on-target delivery, could be used in co-cultures or organoids to deliver transgenes to specific cell types, enabling refined studies of these systems. For in vivo applications, pAG may not be suitable due to the competition of antibodies in serum, but the scFv and SNAP strategies can be deployed in these situations, although reducing the off-target uptake of DIRECTED particles by macrophages will be required to achieve efficient delivery in vivo.

Our exploration of alternative envelopes revealed that baculoviral GP64 from *Autographa californica* multiple nucleopolyhedrovirus (AcMNPV) is compatible with DIRECTED. The availability of multiple fusogenic components could be essential for the re-dosing of patients with transient genetic medicines or for cases of pre-existing immunity. It is also tempting to speculate that the incorporation of additional targeting ligands (e.g., using the pAG or SNAP strategy) into baculoviruses would increase their efficiency for delivery to mammalian cells. We also showed that DIRECTED works with lentiviral vectors, CreVLPs, and eVLPs to deliver integrating genetic payloads, proteins, and RNPs, respectively. Furthermore, it is likely that DIRECTED can also be used

with other enveloped delivery modalities, such as exosomes or retroviruses. Importantly, by increasing on-target delivery and decreasing off-target delivery, DIRECTED could lead to reductions in the required dose of these delivery vehicles, thereby reducing negative side effects such as liver toxicity[56]. In line with this, our in vivo experiments show lower delivery efficiency to monocytes and B cells for CD5-DIRECTED particles when compared to CreVLPs pseudotyped with wild-type VSV-G.

The generation of CAR-T cells relies on the purification and ex vivo transduction of T cells from patients, which limits the accessibility of the treatment. This is, at least in part, due to the long manufacturing process and the high cost associated with the production of CAR-T cells[57,58]. We show that SNAP-DIRECTED particles are capable of transducing non-activated T cells, which could reduce the time and resources needed for clinical CAR-T generation by coupling cellular activation with transduction. Importantly, the modular nature of DIRECTED components enables the incorporation of an additional targeting moiety in existing protocols without major changes. Moreover, the generation of CAR-T cells in vivo may even further increase accessibility to these therapies. To this end, the transduction efficiencies that we observed in whole blood were in the same range as previously reported values for VSV-G and engineered CD3-targeting viruses in whole blood[59]. These levels are higher than the vaccine-induced expansion of individual T cell receptor clones, which typically reach about 0.2% of all CD4 T cells[60,61]. The level of transduction that we achieved falls within the range of specific T cell clones in the acute phase of viral infections, which can reach up to 2%[62]. Thus, the achieved transduction efficiency does represent a biologically meaningful proportion of T cells, which in the case of in vivo CAR-T therapy, would further expand upon antigen encounter and could enable the generation of CAR-T cells in vivo using a single-step protocol.

Through our experiments, we identified several principles for choosing cell surface receptors that are compatible with DIRECTED particles: (1) higher cell surface expression level of a receptor favors more effective delivery; (2) the targeted receptor needs to undergo endocytosis to enable the DIRECTED particles to get to the endosomal compartment and experience low pH, where the fusogen becomes activated; and (3) to increase the set of potential receptors, the SNAP strategy affords the most flexibility, presumably due to the reduced complexity of virus-receptor binding as the targeting antibody and virus are covalently linked. However, the use of NHS ester chemistry for the functionalization of antibodies with benzylguanine, necessitates individual optimization of the labeling step. Future research will be needed to determine the quantitative relationship between the biophysical properties of receptors, such as internalization rate, and the success of transgene delivery. In line with this, endogenous ligands or agonistic antibodies that increase receptor internalization may boost delivery efficiency. Future studies to optimize the efficiency of antibody-tethering on the virion surface, evaluate additional mutations that could help to reduce off-target delivery, and identify receptors that allow specific targeting of unique cell types in vitro and in vivo will further advance the DIRECTED technology and help to address the critical unmet need for cell type-specific delivery of molecular cargoes.

In summary, the presented data suggest that SNAP-DIRECTED particles allow the integration of multiple targeting ligands, as well as agonistic molecules, to achieve cell type-specific delivery. This approach potentially enables the manipulation of cell fate and cell state of specific target cells in situ and may allow for the delivery of therapeutic payloads, such as replacement genes or genome editing tools, both in a transient and integrating manner.

# Methods

## Ethical statement
All experiments were performed in compliance with all relevant ethical regulations as approved by the Institutional Biosafety Committee (IBC) of the Broad Institute (Protocol # IBC-2017-00146).

## Cloning of constructs
Gibson assembly was carried out using Gibson assembly Master Mix (NEB, Cat. # E2611L) in 10-μl reaction volume using 50 ng of backbone per reaction and a threefold molar excess of other fragments. Reactions were incubated for at least 1 h at 50 °C before 2 μl were transformed into chemically competent Stbl3 cells (Thermo Fisher Scientific, Cat. # C737303) and plated on agar plates containing the corresponding antibiotic for selection (Concentrations: Ampicillin 100 μg/ml; Kanamycin 50 μg/ml). The next day single colonies were picked into TB containing the corresponding antibiotic (Concentrations: Ampicillin 100 μg/ml; Kanamycin 50 μg/ml) and allowed to grow overnight before plasmids were extracted by miniprep (Qiagen, Cat. # 27106 × 4). All plasmid sequences were subsequently confirmed by sequencing.

Site-directed mutagenesis was carried out using a commercial site-directed mutagenesis kit (NEB, Cat. # E0554S) according to the manufacturer's instructions, with the following change: Polymerase Chain Reactions were performed using Phusion Flash Polymerase (Thermo Fisher Scientific, Cat. # F548L) or KOD Xtreme polymerase (Millipore Sigma, Cat. # 71975-3). Plasmids that were used for virus production were purified by MidiPrep to ensure high quality (Zymo, Cat. # D4201). Base constructs are listed in Table 1. An overview of all constructs generated in this study (which are also deposited on addgene) and the oligonucleotides used can be found in Supplementary Data 2.

## Antibodies
All antibodies used in this study can be found in Supplementary Data 3.

BG-labeled antibodies were used at the same dilutions as unlabeled antibodies for DIRECTED tropism.

## Cell culture
HEK293FT cells were purchased from Thermo Fisher Scientific (Cat. # R70007) and maintained in DMEM (Thermo Fisher Scientific, Cat. # 10569044) supplemented with 10% FBS (Seradigm, Cat. # 1500-500) and 1x Penicillin and Streptomycin (Pen/Strep; Thermo Fisher Scientific, Cat. # 15140163). Jurkat E6 cells were purchased from ATCC (Cat. # TIB152) and maintained in RPMI-1640 (Thermo Fisher Scientific, Cat. # 61870127) supplemented with 10% FBS and 1x Pen/Strep. A549+Ace2 cells were a gift from Ben Gewurz[63] (initially received from ATCC Cat. # CCL-185) and maintained in RPMI-1640 supplemented with 10% FBS and 1x Pen/Strep. Kasumi-1 cells were purchased from ATCC (Cat. # CRL-2724) and maintained in RPMI-1640 with 20% FBS and 1x Pen/Strep. OUMS-23 cells were purchased from the Genetic Perturbation Platform (GPP) at the Broad and maintained in DMEM supplemented with 10% FBS and 1x Pen/Strep. HepG2 cells were purchased from ATCC (Cat. # HB-8065) and maintained in EMEM (ATCC, Cat. # 30-2003) supplemented with 10% FBS and 1x Pen/Strep. K562 + HLA-A2+eGFP cells were maintained in RPMI-1640 supplemented with 10% FBS and 1x Pen/Strep. These cells were a gift from Prof. Baltimore[64] and initially received from ATCC (Cat. # CRL-3343). All cell lines were confirmed to be mycoplasma free. All cells were cultured in a humidified incubator at 37 °C and 5% $CO_2$. FBS was heat inactivated before use for 30 min at 56 °C.

## Isolation of PBMCs from buffy coats
Buffy coat preparations containing peripheral blood mononuclear cells (PBMCs) were diluted 1:1 in preparation buffer (PBS with 0.1% BSA and 2 mM EDTA), and mononuclear cell separation was performed by density centrifugation (Ficoll-Paque Premium 1.084, Cytiva) with diluted peripheral blood cells (centrifugation 30 min without brakes, $600 \times g$) in 50-mL Leucosep(R) tubes with porous barrier (Greiner Bio, Cat. # 227290 P). Cells were carefully aspirated and washed with preparation buffer (centrifugation 5 min at $500 \times g$). Red blood cells were lysed using ACK Lysis Buffer (Gibco, Cat. # A1049201) for 1 min at room

**Table 1 | Base constructs**

| Name | Source | Comment | Link |
|---|---|---|---|
| psPAX2 | Addgene plasmid # 12260 | Gift from Didier Trono | http://n2t.net/addgene:12260 |
| MD2.G | Addgene plasmid # 12259 | Gift from Didier Trono | http://n2t.net/addgene:12259 |
| pCMV-15F11-HA-Halo | Addgene plasmid # 129592 | Gift from Tim Stasevich[66] | http://n2t.net/addgene:129592 |
| RV-Cag-Dio-GFP | Addgene plasmid # 87662 | Gift from Oscar Marin[67] | http://n2t.net/addgene:87662 |
| pME-puro-SNAP-FLAG-CD59 | Addgene plasmid # 50374 | Gift from Reika Watanabe[68] | http://n2t.net/addgene:50374 |
| pHCMV-AmphoEnv | Addgene plasmid # 15799 | Gift from Miguel Sena-Esteves[69] | http://n2t.net/addgene:15799 |
| pDisplay-mSA-EGFP-TM | Addgene plasmid # 39863 | Gift from Sheldon Park[70] | http://n2t.net/addgene:39863 |
| pSLQ8703 pHR-R(TRE3G-Fluc-2XcrIFNg-SV40pA)-EF1a-sfGFP-T2A-rtta-WPRE | Addgene plasmid # 140230 | Gift from Stanley Qi[71] | http://n2t.net/addgene:140230 |
| m168 | Addgene plasmid # 34886 | Gift from Irvin Chen[24] | http://n2t.net/addgene:34886 |
| MMLVgag-3NES-Cas9-NLS | Addgene plasmid # 181752 | Gift from David Liu[6] | http://n2t.net/addgene:181752 |
| pLVX-EF1alpha-SARS-CoV-2-M-2xStrep-IRES-Puro | Addgene plasmid # 141386 | Gift from Nevan Krogan[72] | http://n2t.net/addgene:141386 |
| pEx1-pEF-H2B-mCherry-T2A-rTetR-KRAB-Zeo | Addgene plasmid # 78352 | Gift from Michael Elowitz[73] | http://n2t.net/addgene:78352 |
| lenti dCAS-VP64_Blast | Addgene plasmid # 61425 | 74 | http://n2t.net/addgene:61425 |
| pLenti CMV rtTA3 Blast (w756-1) | Addgene plasmid # 26429 | Gift from Eric Campeau | http://n2t.net/addgene:26429 |
| plentiGuide-Puro | Addgene plasmid # 52963 | 75 | http://n2t.net/addgene:52963 |

temperature, and cells were washed (centrifugation 5 min, $500 \times g$) and resuspended in preparation buffer. After cell counting, $1 \times 10e7$ cells were frozen per aliquot in 90% FCS (VWR, Cat. # 97068-085) supplemented with 10% DMSO (Sigma-Aldrich, Cat. # D2438-5X10ML) and stored in liquid nitrogen until further use.

**Lentiviral vector production**
On the day before transfection, HEK293FT cells were seeded at a cell density of $8.3E4$ cells per cm² in DMEM with 10% FBS and 1x Pen/Strep. The next day, cells were transfected with a mix of envelope plasmid (27%), the lentiviral helper plasmid (29%), and a plasmid encoding the desired transfer genome (44%) using PEI (Polysciences, Cat. # 24765). Whenever applicable, the amount of envelope plasmid was split to account for the envelope and the helper envelope (i.e., protein AG, SNAP, or scFv) without changing the amounts of the lentiviral helper plasmid (29%) or the transfer genome (44%). Four hours after transfection, the media was exchanged for fresh DMEM with 10% FBS and Pen/Strep. Forty-eight hours after transfection, the viral supernatant was harvested, spun at $1500 \times g$ to pellet cell debris, and filtered through a 0.45-μm filter (Millipore Sigma, Cat. # SE1M003M00). Whenever needed, the viral supernatant was concentrated by ultracentrifugation at $26,000 \times rpm$ (~$90,000 \times g$ at $r_{average}$) for 2 h at 18 °C over 2 ml of a 20% sucrose cushion in an AH-629 or SW 28 rotor. The pellet was then resuspended in 1/50th to 1/200th of the initial volume of PBS before being used for titration. To remove excess antibody from antibody-BG SNAP click reactions (performed with 250× concentrated virus), the mixture was resuspended in 25 ml of PBS before being concentrated by ultracentrifugation at $26,000 \times rpm$ (~$90,000 \times g$ at $r_{average}$) for 2 h at 18 °C over 2 ml of a 20% sucrose cushion in an AH-629 or SW 28 rotor.

**Titration of lentiviral particle number by RT-qPCR**
To determine the number of viral genomes in lentiviral vector preps, the supernatant or concentrated viral stock was incubated with benzonase (Sigma-Aldrich, Cat. # E1014-25KU) to remove free nucleic acids, such as plasmids and RNA resulting from cell death, by incubating up to 100 μl of viral vector suspension with 0.5 μl of benzonase and 0.4 μl 1 M magnesium chloride (Thermo Fisher Scientific, Cat. # AM9530G). If the volume of viral vector suspension was lower than 100 μl, the rest of the volume was filled up with ultrapure water (Thermo Fisher Scientific, Cat. # 10977015). Subsequently, the reaction was incubated for at least 2 h at 37 °C before viral RNA was extracted using a viral RNA extraction kit (Zymo, Cat. # R1041 or R1035) according to the manufacturer's instructions. The viral genome copy number was determined using an RT-qPCR kit (Lenti-X Quant, Takara, Cat. # 631235). All reactions were run on a BioRad CFX Opus 384 RT-PCR machine (Cat. # 12011452).

**Transduction of cell lines in vitro**
Target cells were seeded at 5000 or 10,000 cells per well of a 96-well plate and directly incubated with viral particles at the indicated multiplicities of infection (MOI) in their cognate media (in the presence of FBS). Twenty-four hours later, the media was exchanged with fresh media, and cells were incubated for at least another 72 h before analysis. For transduction enhancer tests, 8 μg/ml of Polybrene (Santa Cruz Biotechnology Cat. # sc-134220) or 10 μg/ml of vectofusin (Miltenyi Biotec, Cat. # 130-111-163) were added. Spinoculations were performed at $1500 \times g$ for 90 min at 33 °C. For DIRECTED tropism, the indicated antibody amounts were used unless otherwise stated. For the in vitro transduction of Kasumi-1 cells, lentiviral particles were co-incubated with αCD117 or αCD20 antibody for 30 min at room temperature, before excess antibody was removed using Amicon Ultra-0.5 Centrifugal Filter Units (Millipore Sigma, Cat. # UFC510096) by washing three times with excess PBS.

**Transduction of PBMCs with T cell and B cell targeting DIRECTED lentiviral vectors**
**T cell targeting (Supplementary Fig. 6a).** For T cell targeting in purified PBMCs, 150,000 cells were seeded per well of a 96-well plate in TCM with 50U/ml rhIL-2 (PeproTech Cat. # 200-02) on the day before transduction. DIRECTED particles were prepared by incubating 80 μl of 100x concentrated VSV-Gdm+SNAP with either 6 μg αCD3-BG (VSV-Gdm+αCD3), 2.5 μg αCD3, 1.25 μg αCD28, and 2.5 μg αCD4 (VSV-Gdm-Tcell), or 2.5 μg αCD19-BG and 0.5 μg MegaCD40L-BG (VSV-Gdm-Bcell) overnight at 4 °C. For DIRECTED particles with expanded tropism, 80 μl of 100x concentrated VSV-G+SNAP were incubated with 6μg αCD3 overnight at 4 °C. The next day unreacted antibody was removed by harvesting the functionalized particles by ultracentrifugation through a 20% sucrose cushion, and the pellet was resuspended in 80 μl of PBS. VSV-Gdm+SNAP and VSV-G were used without additional purification. Next, media on cells was exchanged with fresh TCM with 50U/ml rhIL-2, 8 μg/ml Polybrene (Santa Cruz Biotechnology Cat. # sc-134220) and 2 μl per well of Fc block (BioLegend Cat. # 422302). Fifteen minutes later virus was added at an MOI of 500 (for VSV-Gdm+SNAP and VSV-G) or 250 (for VSV-Gdm-Bcell, VSV-Gdm-Tcell, VSV-Gdm-αCD3) in two wells each and cells were spinoculated for 1.5 h at $1000 \times g$ and 33 °C. After overnight incubation, media was exchanged

with fresh TCM (supplemented with 50U/ml rhIL-2) and cells were incubated for an additional 5 days with a media change at day 3 post-transduction. On day 6, cells were collected and half of the cells were used for analysis by flow cytometry, while the other half was used to stimulate T cell outgrowth by adding αCD3 and αCD28 T cell activator beads (Thermo Scientific Cat. # 11131D) in fresh T cell media supplemented with 50U/ml rhIL-2, 25U/ml rhIL-7 (PeproTech Cat. # 200-07), and 50U/ml rhIL-15 (PeproTech Cat. # 200-15). Media was exchanged every other day, and cells were debeaded and analyzed by flow cytometry on day 14 post-infection.

### B cell targeting (Supplementary Fig. 6f).
DIRECTED particles were prepared by incubating 80 µl of 100x concentrated VSV-Gdm+SNAP or VSV+SNAP with 2.5 µg αCD19-BG and 0.5 µg MegaCD40L-BG (VSV-Gdm-Bcell, or VSV-G-Bcell) overnight at 4 °C. The next day unreacted antibody was removed by harvesting the functionalized particles by ultracentrifugation through a 20% sucrose cushion, and the pellet was resuspended in 80 ml of PBS. VSV-Gdm+SNAP, VSV-G+SNAP, and VSV-G were used without additional purification. Then, 100,000 freshly thawed PBMCs were seeded per well of a 96-well plate in TCM with 50U/ml rhIL-2 and 8 µg/ml Polybrene and incubated with viral vectors at an MOI of 3500 for VSV-Gdm, VSV-G+SNAP, or VSV-G and an MOI of 1750 for VSV-Gdm-Bcell or VSV-G-Bcell in two wells each and cells were spinoculated for 1.5 h at 1000 × *g* and 33 °C. Right after spinoculation, 2 µl of αCD3 and αCD28 T cell activator beads were added per well. After overnight incubation, media was exchanged with fresh TCM (supplemented with 50U/ml rhIL-2, 25U/ml rhIL-7, and 50U/ml rhIL-15), and cells were incubated for an additional 5 days with a media change at day 3 post-transduction.

### Transduction of cells in whole blood (Supplementary Fig. 7a)
Fresh blood from two human donors was obtained from Research Blood Components in CPT-NaCitrate tubes. Directly upon arrival, 100 µl of blood was transferred to a 96-well deep-well plate and incubated with 2 µl of Fc block (BioLegend Cat. # 422302) per well. Thirty minutes later, viral vectors (prepared as described in "Transduction of PBMCs with T cell and B cell targeting DIRECTED lentiviral vectors") were added at an MOI of 1000 for VSV-Gdm and VSV-G or at an MOI of 500 for VSV-Gdm-Tcell and VSV-Gdm-αCD3. MOIs were calculated based on an assumption of 5000 leukocytes per µl blood. The cells were incubated for 6 h shaking in a humidified incubator at 37 °C and 5% CO$_2$. After this time, red blood cells were lysed using RBC lysis buffer (BioLegend, Cat. # 420301), and the cells (containing granulocytes, monocytes and leukocytes) were transferred to TCM (supplemented with 50U/ml rhIL-2, 25U/ml rhIL-7, and 50U/ml rhIL-15), and 2 µl of T cell activator beads were added per well. The next day media was replaced with fresh TCM (supplemented with 50U/ml rhIL-2, 25U/ml rhIL-7, and 50U/ml rhIL-15), and cells were incubated for another 5 days with media changes every other day. On day 6, cells were debeaded, washed, and half of the cells were incubated with fresh T cell activator beads (2 µl per well), whereas the other half was used for analysis. Media was exchanged every other day on the cultured cells. The remaining cells were debeaded, washed and analyzed on day 14 post-transfection.

### Stable cell line generation using lentiviral vectors
To establish HEK293FT surface-HA, HEK293FT surface-Spot, HEK293FT surface-Strep, and HEK293FT surface-V5 cell lines, cells were seeded at 50,000 cells/well in 6-well plates and transduced with 1 ml of viral supernatant. Twenty-four hours later, media was exchanged for fresh media, and the next day, antibiotic selection was started using 1 µg/ml Puromycin (Thermo Fisher Scientific, Cat. # A1113803). Cells were not maintained under Puromycin selection for in vitro infection experiments.

To establish the Jurkat+surface-HA cell line, Jurkat E6 wild-type cells were seeded at 5E5 cells/ml and incubated with 3 ml of viral

supernatant in 10 ml total media. Twenty-four hours later, media was exchanged for fresh media, and the next day antibiotic selection was started using 1 µg/ml Puromycin (Thermo Fisher Scientific, Cat. # A1113803). Cells were not maintained under Puromycin selection for in vitro infection experiments.

To establish a Jurkat E6 Cre Reporter cell line, 5E5 cells/ml wild-type Jurkat E6 cells were transduced with 3 ml lentiviral supernatant containing a Cre-responsive GFP reporter (pBS-loxP-GFP) in 10 ml total media. The next day media was exchanged for fresh media, and the day after antibiotic selection with Blasticidin S HCl (Thermo Fisher Scientific, A1113903) was started at 10 µg/ml. Cells were selected for 2 weeks, and surviving cells were plated by limited dilution into 96-well plates to grow out clones. Media was exchanged every 2 days, and after 3 weeks, 96-well plates were split in half, where one plate was tested using 100 µl of VSV-G CreVLPs to identify clones that show high switching frequency. The clone showing the highest switching capacity (-90%) was expanded and used for in vitro DIRECTED-CreVLP experiments.

To establish HEK293FT ΔB2M cells, wild-type cells were seeded at 50,000 cells/well in 6-well plates before treating them with 1 ml of VSV-G pseudotyped eVLPs delivering Cas9-RNPs with a B2M targeting sgRNA. The cells were subsequently expanded and evaluated for the loss of B2M surface expression by flow cytometry.

### Cell sorting
HEK293FT ΔB2M cells were stained with αB2M-FITC antibody (BioLegend) at a dilution of 1:100 for 45 min at 4 °C. Subsequently, cells were washed twice and sorted for B2M negative cells on a Sony SH800 sorter. Jurkat E6 + surface-HA cells were stained with aHA-PB450 (1:100) for 45 min at 4 °C, washed twice with Flow buffer (PBS +2% FBS+5 mM EDTA) and sorted into four bins of different expression levels on a Sony MA900 sorter.

### Benzylguanine labeling of antibodies
Antibodies were desalted using Zebaspin Desalting Columns (7 kDa MWCO, Thermo Fisher Scientific, Cat. # 89882) and diluted to a concentration of 0.5 mg/ml, before incubation with a 40-fold molar excess of BG-GLA-NHS (New England Biolabs, Cat. # S9151S). To test different ratios of BG-GLA-NHS, 10-, 20-, 40-, or 80-fold molar excess of BG-GLA-NHS was added to the antibodies, The reaction was allowed to continue for 30 min at room temperature before it was stopped by removing the unreacted NHS-BG substrate using Zebaspin Desalting Columns (7 kDa MWCO). MegaCD40L (Enzo Lifesciences, Cat. # ALX-522-110-C010) was labeled with a 10-fold molar excess of NHS-GLA-BG, due to the high number of surface exposed lysines. αCD3 (BioLegend Cat. # 317302), αCD4 (BioLegend Cat. # 317402) and αCD19 (BioLegend Cat. # 302202) were each labeled with a 40-fold molar excess of NHS-GLA-BG. αCD28 (BioLegend Cat. # 302933) was labeled with a 20-fold molar excess of NHS-GLA-BG, due to significant precipitation at 40-fold molar excess. All BG-labeled antibodies were resuspended in PBS, and their concentration was determined using the IgG function on a Nanodrop 2000 device.

### Evaluation of benzylguanine labeling efficiency
BG-antibodies were incubated with a twofold molar excess of purified SNAP protein (NEB, Cat. # P9312S) for at least 2 h at 37 °C. The samples were mixed with Bolt Sample Buffer (Thermo Fisher Scientific, Cat. # B0008) supplemented with reducing agent (Thermo Fisher Scientific, Cat. # B0009) and boiled for 5 min at 95 °C before loading onto Bolt 4-12% Bis-Tris gradient gels (Thermo Fisher Scientific, Cat. # NW04125BOX) before proceeding to western blot analysis.

### Identification of homologous proteins and phylogenetic tree generation
Seed sequences were used for Position-Specific Iterated BLAST searches using the NCBI BLAST tool with the following deviations from

standard parameters: Max Target Sequences = 20000, Word Size = 2 and no filtering. After a maximum of 8 iterations, if the search did not converge earlier, the hits were manually verified, and only sequences associated with viruses were kept. The search did not converge after eight iterations of PSIBLAST for VSV-G and converged after six iterations for DHOV_GP. The sequences of hit proteins were downloaded as FASTA files, and sequences with homology of more than 95% were removed using CD-HIT (http://weizhong-lab.ucsd.edu/cdhit_suite/cgi-bin/index.cgi?cmd=cd-hit). Afterward, sequences were imported to Geneious Prime (https://www.geneious.com/prime/) and partial sequences, as well as sequences with sequence ambiguities, were removed manually. Next, the sequences were aligned using the MAFFT Alignment plugin in Geneious Prime with the L-INS-i Algorithm and standard parameters. Finally, the resulting sequence list was used with the fasttree plugin to generate a phylogenetic tree, which was visualized and annotated using iTOL[65] (https://itol.embl.de/).

DHOV_GP
MDSTIRLVATIFLISLTQQIEVCNKAQQQGPYTLVDYQEKPLNISRIQI
KVVKTSVATKGLNFHIGYRAVWRSYCYNGGSLDKNTGCYNDLIPKSPTE
SELRTWSKSQKCCTGPDAVDAWGSDARICWAEWKMELCHTAKELKKYS
NNNHFAYHTCNLSWRCGLKSTHIEVRLQASGGLVSMVAVMPNGTLIPIEG
TRPTYWTEDSFAYLYDPAGTEKKTESTFLWCFKEHIRPTTELSGAVYDTH
YLGGTYDKNPQFNYYCRDNGYYFELPANRLVCLPTSCYKREGAIVNTMHP
DTWKVSEKLHSASQFDVNNVVHSLVYETEGLRLALSQLDHRFATLSRLFN
RLTQSLAKIDDRLLGTLLGQDVSSKFISPTKFMLSPCLSTPEGDSNCHNH
SIYRDGRWVHNSDPTQCFSLSKSQPVDLYSFKELWLPQLLDVNVEGVVA-
DEEGWSFVAQSKQALIDTMTYTKNGGKGTSLEDVLGYPSGWINGKLQG
LLLNGAISWVVVIGVVLVGVCLMRRVF.

## Flow cytometry

All flow cytometry experiments with fluorescent proteins were performed as follows: For adherent cells, media was initially removed, and cells were washed once with PBS (Thermo Fisher Scientific, Cat. # 10010049) before they were released from the dish using TrypLE (Thermo Fisher Scientific, Cat. # 12604021). After 5 min, the reaction was stopped by resuspending the cells in DMEM with 10% FBS and Pen/Strep, and the cell suspension was transferred to a V-bottom 96-well plate before harvesting the cells by centrifugation ($1000 \times g$, 3 min). The cells were washed twice with Flow Buffer (PBS+2% FBS+5 mM EDTA) containing DAPI (100 ng/ml; Thermo Fisher Scientific, Cat. # D1306) before resuspending in Flow Buffer (without DAPI) and analyzed on a Beckman Coulter CytoFLEX S device. Suspension cells were triturated to break up cell clumps before immediately being transferred to a V-bottom plate and harvested by centrifugation ($1000 \times g$, 3 min). As above, cells were washed twice with Flow Buffer containing 100 ng/ml DAPI and finally resuspended in Flow Buffer before being analyzed on a CytoFLEX S device. All data was acquired using CytExpert software and analyzed using FlowJo (https://www.flowjo.com/).

For staining of surface receptors, cells were treated as described above, with an additional incubation of cells with an antibody after the first wash in the Flow Buffer. The used antibodies and corresponding dilutions are listed in the table of antibodies. In the case of fluorophore spillover, unstained and single-stained samples were acquired and used to calculate the necessary compensation in FlowJo. For the flow cytometry analysis of PBMCs, whole blood, or murine splenocytes cell suspensions were incubated with Fc block (for human: BioLegend Cat. # 422302; for mouse: BD Biosciences Cat. # 553142) for 15 min at room temperature before proceeding with the antibody stainings as described above. Instead of DAPI, fixable viability dye 780 (Thermo Scientific Cat. # 65-0865-18) was used and staining was performed in PBS.

## Imaging of HEK293FT cells after transfection with pHCMV-VSVgSS-pAG-TM

8E5 HEK293FT cells were seeded in 6-well plates the day before transfection. The next day cells were transfected with pHCMV-VSVgSS-

pAG-TM (3 µg DNA). Two days after transfection, cells were incubated with a mouse-IgG-FITC antibody (BioLegend, Cat. # 407406) at a dilution of 1:100 for 15 min before washing the cells once in PBS and immediately imaging the cells on an EVOS M5000 system.

## Imaging of transduction efficiency

HEK293FT cells, engineered to express synthetic receptors, were seeded at 10,000 cells/well in a black 96-well plate and transduced with lentiviral vectors co-decorated with protein AG or SNAP in the presence of the indicated antibodies (at 0.25 µg/well). The next day, media was exchanged for fresh DMEM with 10% FBS and Pen/Strep and incubated for another 2 days, before the cells were imaged on an EVOS M5000 microscope. Images were subsequently cropped and adjusted to the same intensity range using Fiji (https://imagej.net/software/fiji/).

## Protein structure prediction using AlphaFold2

Protein structure was predicted using AlphaFold2 via the Google Colab repository (https://github.com/sokrypton/ColabFold/ and https://colab.research.google.com/github/sokrypton/ColabFold/blob/main/beta/AlphaFold2_advanced.ipynb). The algorithm was run using the protein sequences with the following parameters deviating from the standard options: Sampling options; num_models: 3; max_recycles: 12.

## Production of VSV-G competitor

The VSV-G competitor (recombinant diCR2 protein) was produced by transfection of 30 ml Expi-HEK suspension cells with pHCMV-diCR2 using a commercial transfection kit (Thermo Fisher Scientific, Cat. # A14524) according to the manufacturer's instructions. Cells were allowed to produce protein for 5 days before media was collected, filtered through a 0.45-µm filter (Millipore Sigma, Cat. # SE1M003M00), and crude supernatant was stored at 4 °C until use (up to 1 week).

## Production of eVLPs and CreVLPs

On the day before transfection, HEK293FT cells were seeded at a cell density of 8.3E4 cells per cm² in DMEM with 10% FBS and Pen/Strep. The next day, cells were transfected using PEI with the following plasmids: for eVLPs, containing Cas9 and a sgRNA, envelope plasmid (10%), retroviral helper plasmid containing MLV gagpol (35%), plasmid encoding gag-3NES-Cas9 (10%), and plasmid containing an sgRNA expression cassette (45%); for CreVLPs containing Cre recombinase, envelope plasmid (20%), retroviral helper plasmid containing MLV gagpol (60%), and plasmid encoding gag-3NES-Cre (20%). To generate DIRECTED-eVLPs or DIRECTED-CreVLPs, the amount of envelope plasmid was split to account for the envelope (i.e., VSV-G, VSV-G K47Q/R354Q) and the helper envelope (i.e., protein AG, SNAP, or scFv) without changing the amounts of the other plasmids. Four hours after transfection, the media was exchanged for fresh DMEM with 10% FBS and Pen/Strep. Forty-eight hours after transfection, the viral supernatant was harvested, spun at $1500 \times g$ to pellet cell debris, and filtered through a 0.45-µm filter. Whenever needed, the viral supernatant was concentrated by ultracentrifugation at 26,000×rpm (~90,000 × g at $r_{average}$) for 2 h at 18 °C over 2 ml of a 20% sucrose cushion in an AH-629 or SW 28 rotor. The pellet was then resuspended in PBS at 1/50th to 1/200th of the initial volume.

## Western blot

Protein samples were diluted with 4x Bolt Sample Buffer (Thermo Fisher Scientific, Cat. # B0008) supplemented with reducing agent (Thermo Fisher Scientific, Cat. # B0009) before boiling at 95 °C for 5 min. After boiling, samples were allowed to cool to room temperature before being loaded onto Bolt 4-12% Bis-Tris gradient gels (Thermo Fisher Scientific, Cat. # NW04125BOX). Gels were run at 200 V for 24 min in an MES running buffer (Thermo Fisher Scientific, Cat. # B0002). Subsequently, gels were rinsed in distilled water before transfer onto a PVDF membrane (Thermo Fisher Scientific, Cat. #

IB24001) using Program 3 (20 V, 7 min) on an iBlot2 device (Thermo Fisher Scientific, Cat. # IB21001). After transfer, the PVDF membrane was blocked using TBS-T Blocking Buffer (Licor, Cat. # 927-60001) for at least 30 min at room temperature, before the addition of primary antibody and incubation at 4 °C overnight on a shaker device. The next day, blots were rinsed twice with TBS-T (Thermo Fisher Scientific, Cat. # 28360) and then incubated with secondary fluorescently labeled antibody (LiCor, Cat. # 926-68020, 926-32219, or 926-32211) at a 1:2500 to 1:10,000 dilution for at least 45 min at room temperature. Finally, blots were washed three times with PBS-T for at least 5 min per wash before being imaged on a BioRad ChemiDoc Imaging System. Images were exported as raw tif files for all subsequent analyses. All densitometry analyses were performed in Fiji software.

### In vitro Cre recombinase assay
For the in vitro Cre recombinase assays, different amounts of purified Cre recombinase (NEB, Cat. # M0298S) or dilutions of CreVLPs were incubated with pLox2+ (25 ng/reaction, NEB, Cat. # N0416SVIAL) in Cre reaction buffer supplemented with 0.25% NP-40 (Thermo Fisher Scientific, Cat. # 85124) in a total reaction volume of 5 μl. To test the background levels of Cre recombinase in CreVLP preparation, the NP-40 was omitted. Reactions were incubated for 45 min at 37 °C before heat inaction (70 °C, 10 min). Thereafter, samples were diluted 1:10 in ultrapure water, and reactions were run on an Agilent TapeStation device using D5000 ScreenTape (Agilent, Cat. # 5067-5593).

### Cell viability assay
HEK293FT + surface-Spot were seeded at 10,000 cells per well of a 96-well plate in DMEM+10%FBS+Pen/Strep before transduction with lentiviral vectors in the presence or absence of antibody at different MOIs. The next day media was exchanged. On day 4 post-transduction, cells were trypsinized, washed once in PBS and stained with Fixable Viability Dye eFluor 780 (Thermo Scientific, Cat. #65-0865-18) for 30 min at 4 °C. After this incubation, cells were washed and stained with Annexin V-PB450 (BioLegend, Cat. # 640926) for 15 min at room temperature in Annexin V binding buffer according to the manufacturer's instructions. Thereafter an equal volume of Annexin V binding buffer was added, and cells were immediately analyzed by flow cytometry on a Beckman Coulter CytoFLEX S device.

Jurkat E6 cells were seeded at 25,000 cells per well in 50 μl per well in a 96-well plate in RPMI+10%FBS+Pen/Strep + 8 μg/ml Polybrene (Santa Cruz Biotechnology Cat. # sc-134220). Subsequently, the cells were transduced with lentiviral vectors at the indicated MOIs. αCD3-SNAP-DIRECTED particles were purified by ultracentrifugation through a sucrose cushion as described above to remove excess unreacted antibody. The next day, media was exchanged, and cells were resuspended in 100 μl RPMI+10%FBS+Pen/Strep. On day 4, cells were stained as described above and immediately analyzed by flow cytometry on a Beckman Coulter CytoFLEX S device.

### Animal experiments
All animal experiments were approved by the Institutional Animal Care and Use Committee (IACUC) of the Broad Institute (Protocol ID 0017-09-14-2). Animal maintenance complied with all relevant ethical regulations and were consistent with local, state and federal regulations as applicable, including the National Institutes of Health Guide for the Care and Use of Laboratory Animals. Animals were kept on a 12-h light/dark cycle between 68 °F and 79 °F and 30–70% humidity. Mice had ad libitum access to standard rodent diet and water. 4–6-week-old female Ai14 mice (N = 17) were ordered from Jackson Laboratories (Strain # 007914) and injected with CreVLPs via tail vein injection. Each animal received ~50 μl of the purified CreVLP suspension. Four animals were injected with VSV-Gdm+SNAP+αCD5-BG CreVLPs (αCD5), five animals with VSV-Gdm+SNAP CreVLPs (no Antibody), and six animals were injected with VSV-G CreVLPs (VSV-G). Five days post-injection, animals

were euthanized by $CO_2$ asphyxiation and cervical dislocation before perfusion with 25 ml PBS by cardiac puncture. Subsequently, spleens were isolated by dissection.

### Analysis of tdTomato+ cells in spleen
Half of the spleens of the animals were used to isolate cells as follows: Spleens were minced through a 70-μm cell strainer, and the cell strainers were washed with 15 ml of PBS, then cells were collected by centrifugation (500 × g, 5 min). The cell pellets were resuspended in 10 ml of RBC lysis buffer (BioLegend Cat. # 420301), and RBC lysis was performed for 5 min at room temperature before the reaction was quenched with 10 ml of PBS. Cells were pelleted and washed twice before proceeding with the blocking of Fc receptors using Fc block (BD Biosciences, Cat. # 553142) for 10 min at room temperature. Cells were collected by centrifugation, washed once in PBS, and resuspended in 5 ml of PBS. Then, 100-μl aliquots of the cell suspension were stained with 2 μl of αCD3-BV421 (BioLegend Cat. # 100228), αCD20-BV421 (BioLegend Cat. # 150405), or αCD11b-BV421 (BioLegend Cat. # 101236) and fixable viability dye eFluor 780 (1:1000, Thermo Scientific Cat. # 65-0865-18) in individual wells of a 96-well plate for 30 min at 4 °C. After the incubation period, 100 μl of fresh PBS was added to the wells, cells were collected by centrifugation and washed twice with 200 μl PBS. Finally, cells were resuspended in 100 μl Flow Buffer (PBS+2% FBS+5 mM EDTA) and analyzed on a Beckman Coulter CytoFLEX S device.

### Immunofluorescence analysis of spleens
The second halves of the spleens and livers were fixed in 4% PFA (Electron Microscopy Sciences Cat. # 157-4) overnight at 4 °C, before being transferred into a 30% sucrose solution for dehydration. After 24 h of dehydration, the spleen fragments were embedded in OCT (Sakura Finetek USA Inc Cat. # 4583) and stored at −80 °C until processing. To cut sections, the molds were equilibrated to −20 °C in a Cryostat device, before being cut into 10-μm slices and immediately mounted onto microscopy slides (VWR Cat. # 48311-703). The slides were allowed to air dry before being transferred to −80 °C until they were stained. The sections were rehydrated and permeabilized with PBS containing 0.5% Triton-X100 (Sigma-Aldrich Cat. # X100-100ML). Next, sections were blocked using 10% donkey serum (Sigma-Aldrich Cat. # D9663-10ML) and Fc block (1:50, BD Biosciences 553142) and stained with an antibody mix (αRFP [1:500, NanoTag Biotechnologies Cat. # N0404], αF4/80-AF647 [1:100, BioLegend Cat. # 123122], and αCD5-FITC [1:100, BioLegend Cat. # 100606]) in PBS-T with 2% donkey serum overnight at 4 °C. The next day, sections were washed (3 times with PBS-T). Subsequently, sections were overlaid with coverslips (VWR Cat. # 48393-221) using DAPI-containing mounting media (Cell Signaling Technology Cat. # 8961S). Coverslips were sealed with nail polish. Images were acquired on a Leica DMI8 Confocal Microscope running Leica Application Suite X (1.4.3), equipped with a Lecia Stellaris 5 camera using an HC PL APO CS2 20x/0.75 DRY objective and a pinhole setting of 1 Airy Unit. Images were processed using Fiji (https://imagej.net/software/fiji/downloads).

### Statistics and reproducibility
No statistical method was used to predetermine the sample size. No data were excluded from the analyses. The experiments were not randomized. The investigators were not blinded to allocation during experiments and outcome assessment. All statistical analyses were performed in R (v 4.2.2). For the analyses presented in Figs. 1b–d; 2b; 3a; 4c, f; 5a–c and Supplementary Figs. 1d–g; 2a, d, h, j, k; 3a; 4a, g; 5f, h a two-sided Welch's t-test with multiple hypothesis correction was performed (either Bonferroni or BH). This was done using the t_test and adjust_pvalue functions from the rstatix package. For the analyses presented in Fig. 2a and Supplementary Fig. 1h, i an ANOVA was performed (aov function from stats package), and if significant, a Dunnett's post-hoc test was used (DunnettTest function from DescTools

package) followed by a Bonferroni correction. For Supplementary Fig. 1l, a linear model was fit to the data using the lm function (stats package). For analyses in Supplementary Fig. 3b, c, a Kruskal test (kruskal.test function from stats package) was performed, which was followed by a Dunnett's post-hoc test if significant. Analyses in Fig. 6c, e, Supplementary Figs. 6a, g, 8a, b used a two-sided Welch's t-test. Analyses in Supplementary Figs. 6e and 7a used a paired, two-sided Welch's t-test. Analyses in Fig. 6b and Supplementary Fig. 6d used a paired, two-sided Welch's t-test with BH correction. All reported test values are provided in the Source Data.

## Reporting summary

Further information on research design is available in the Nature Portfolio Reporting Summary linked to this article.

## Data availability

The NCBI non-redundant protein sequences (nr) database can be found at https://blast.ncbi.nlm.nih.gov/Blast.cgi?PROGRAM=blastp&PAGE_TYPE=BlastSearch&LINK_LOC=blasthome. All other data supporting the findings of this study are available within the paper and its Supplementary Information. Source data are provided with this paper.

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

## Acknowledgements

We thank R. Raghavan and K. DeLong for experimental assistance; B. Lash, G. Edmonds, and K. Kappel for experimental advice; J. Strecker, S. Kannan, S. Zhu, and A. Suberski for help with plasmid sequencing; and all members of the Zhang lab for support and discussions. D.S. was supported by fellowships from the Swiss National Science Foundation (P400PB_199261 and P2ELP3_187926). F.Z. is supported by the Howard Hughes Medical Institute; Milky Way Research Foundation; K. Lisa Yang and Hock E. Tan Molecular Therapeutics Center at MIT; K. Lisa Yang Brain–Body Center at MIT; Broad Institute Programmable Therapeutics Gift Donors; The Pershing Square Foundation, William Ackman, and Neri

Oxman; James and Patricia Poitras; BT Charitable Foundation; Asness Family Foundation; Kenneth C. Griffin, the Phillips family; David Cheng; and Robert Metcalfe.

## Author contributions

D.S. and F.Z. conceived the project and designed experiments. D.S., C.J.F., and M.F. performed experiments. D.S. and G.F. carried out the computational analysis of fusogens. F.Z. supervised the research and experimental design with support from R.M. D.S., R.M., and F.Z. wrote the manuscript with input from all authors.

## Competing interests

D.S. and F.Z. are coinventors on a pending patent application (PCT/US22/52871) related to this work filed by the Broad Institute and MIT. M.J.F. reports receiving speaker's bureau honoraria from Pfizer. F.Z. is a scientific advisor and cofounder of Editas Medicine, Beam Therapeutics, Pairwise Plants, Arbor Biotechnologies, and Aera Therapeutics. F.Z. is a scientific advisor for Octant. The remaining authors declare no competing interests.
