## [Peer Review File · Nature Communications]

Reviewers' Comments:

Reviewer #1:

Remarks to the Author:

This manuscript shows an elegant approach to redirect enveloped viral vectors, in particular pseudotyped lentiviral vectors towards target cells, making use of 3 different approaches to display antibodies or antibody fragments on the surface of the viral envelopes. The strength of this approach is the ability to use commercial antibodies for redirecting and the option for covalent attachment as is the case for the SNAP-tag approach. In addition, the authors show that this REDIRECT approach can work with different fusogens, including VSV-G and mutant versions that lack LDL-R binding.

The ability to redirect viral vectors for cell-type specific transduction is a long held desire and a need for direct in vivo modification of cells (e.g. generation of CAR T cells directly in vivo). Several approaches have been evaluated, including envelope engineering, pseudotyping as well as redirecting lentiviral vectors using bispecific antibodies.

Even though this manuscript nicely shows proof-of-concept for target-cell specific transduction, the redirection of viral vectors is not novel. Several groups have recently shown that this is feasible by using bispecific antibodies (<https://doi.org/10.1128/mBio.02990-19>; <https://doi.org/10.1136/jitc-2021-002737>; <https://doi.org/10.3390/v14102157>). The use of bispecific antibodies has as main advantage that no engineering of the viral envelope is needed, which favours high titres. It would be interesting to learn how these two approaches relate to each other and I therefore suggest a direct comparison.

Furthermore, all experiments presented were performed in vitro on cell types with high expression levels of a target receptor or with artificial receptors introduced by lentiviral transduction. This is rather artificial and what is missing in my opinion are experiments to demonstrate that this REDIRECT approach would also work in a more complex environment, for example cell-type specific transduction of T cells or NK cells in whole blood. This would really demonstrate the value of the REDIRECT approach. As the authors already point out, some of the approaches might not work in vivo due to e.g. inactivation of VSV-G by human serum components, competition of IgG for protA etc. etc. It would be very informative to learn which of the combinations of fusogens and antibody attachment methods would indeed give robust in vivo or ex vivo transduction.

Minor comments:

Statistics are missing for all figures. With only mean values and the individual data points, it is difficult to estimate if some values are significantly different.

"We next sought to block the intrinsic VSV-G tropism through addition of a competitor molecule (Supplementary Figure 1f)" S1F shows HLA-A2 targeted data, not inhibition with a competitor molecule recombinant diCR2 protein.

Unclear if transduction of cells was performed in presence or absence of FBS. Please add info to M&M

Efficiency of transduction with the SNAP-REDIRECTED approach (Fig 2C) seems to be very much dependent on type of antibody. Did the authors test different ratios of the benzylguanine label to protein?

Reviewer #2:

Remarks to the Author:

In this manuscript, Streibinger D et al., describe a modular viral envelope design consisting of separate fusion and targeting moieties that achieve cell specific transgene, protein and RNP delivery. The methodology, dubbed DIRECTED, relies on the co-expression of a viral fusogen

(mutated or not to block its capacity to interact with a specific cellular receptor) and a transmembrane protein which is engineered to bind antibodies and allow users to define the tropism of the produced viral particles.

Using DIRECTED with wild-type VSV-G as a fusogen and a targeting moiety consisting of the VSV-G transmembrane domain fused to pAG, authors produced lentiviral vector particles that were then programmed with an HLA-A2-targeting antibody. Produced particles were able to transduce HEK293FT cells in a partially antibody-dependent manner. Different mutants of VSV-G were then generated to abolish its binding capacity and rely exclusively on the pAG protein to specify targeting. This approach significantly reduced off-target binding in the absence of a bound antibody, while still allowing high lentiviral titers and efficient and specific transduction of target cells in the presence of a targeting antibody. Authors then engineered additional targeting moieties relying on a single chain variable fragment (scFv) or SNAP-tag fused to the VSV-G transmembrane domain which, similarly to pAG, also led to cell-specific targeting, each with variable efficiency depending on the nature of the targeted surface receptor.

Having validated DIRECTED, authors then performed a sequence-based homology search for other fusogens related to VSV-G, that led to the identification of several fusogens from the Rhabdoviridae family (one, Cocal virus G protein, being functional to replace VSV-G). The search for alternative fusogens was then expanded to a library of fusogens from different viral families, which further identified several functional fusogens compatible with the DIRECTED approach. Among these, GP64 from baculovirus performed efficiently, therefore expanding the list of fusogens compatible with DIRECTED that could be specifically useful for in vivo studies where VSV-G is known to be rapidly inactivated by serum proteins.

Finally, authors show that DIRECTED is compatible with retroviral Gag proteins in addition to the lentiviral Gag protein used throughout the previous figures. More particularly, they validated DIRECTED with Cre- and Cas9-VLPs for direct protein and RNP delivery.

Overall the manuscript is of excellent technical quality and tackles one of the last obstacles limiting the use of viral vectors in vivo and in cell types that are difficult to target with unmodified viral envelopes. Although the technical approach is not entirely novel at conceptual level (see point 1 below), the authors go beyond what has been already published regarding the optimization of their platform (although experiments with primary cells or in vivo are missing). Furthermore, the identification of new compatible fusogens (other than the mutant VSV-G) will be of great interest for scientists working in a wide-range of research fields.

Below you will find some major and minor points that, in my opinion, should be addressed by authors in order to improve the manuscript.

1. Line 78-80 : In the introduction, authors mention a recent manuscript (Dobson CS et al., 2022) that describes a VSV-G variant that has lost affinity for LDL-R and which can be used to identify antigen-specific T cells. However, the method described in that manuscript is not restricted to the identification of antigen-specific T cells, but was also used by authors to mediate cell type-specific lentiviral delivery of transgenes in a conceptually similar modular platform as DIRECTED, combining a mutated VSV-G (bearing mutations at the same residues as those selected for the optimal DIRECTED protocol, K47 and R354) with a broad range of targeting moieties. This information should be clearly acknowledged in the manuscript text, for example in lines 114-115 when authors mention the idea of a modular platform. Moreover, there was a second manuscript published in the same issue as Dobson CS et al., that should be cited as well in line 78-80 (<https://doi.org/10.1038/s41592-022-01437-y>).

Finally, the chimeric Sindbis virus envelope m168 that can be conjugated with an antibody similarly to DIRECTED was successfully validated in vivo and shown to specifically target melanoma cells in mice (<https://www.nature.com/articles/nm1192>). This manuscript should also be cited so that readers are aware of the pre-existing literature related to DIRECTED.

2. Authors indicate the MOI (calculated as VGs/cell) used for DIRECTED experiments starting from Figure 1d. However, it will be informative to compare the MOIs obtained with WT VSV-G to those obtained with DIRECTED in order to know the relative efficiency of each setup. It is also important for authors to show cell viability upon transduction with VSV-G pseudotyped particles and when using DIRECTED (one antibody type will be enough) both for HEK293FT and Jurkat cells.

3. Most surface receptors targeted correspond to abundant proteins. It would be interesting for future users to know how well DIRECTED performs when surface receptors are less abundant. Could authors sort Surface-HA cells to obtain populations displaying different HA expression levels and test how well DIRECTED performs at each expression level of the surface receptor?

4. The transmembrane domain of VSV-G used in the targeting module contains residues that are critical for the monomeric to trimeric conformation and subsequent hemi-fusion process (<https://www.nature.com/articles/s41598-018-28868-y>, <https://www.pnas.org/doi/10.1073/pnas.95.7.3425>, <https://www.pnas.org/doi/10.1073/pnas.1618883114>). Could this perturb core VSV-G trimer formation at the hemi-fusion step and explain the decrease in transduction efficiency observed in Figure 1e when adding large amounts of antibodies.

5. Figure 4: Testing DIRECTED with MMLV Gag VLPs was done with the mutant VSV-G fusogen only. Since GP64 is a good candidate for in vivo experiments in order to circumvent VSV-G inactivation by serum proteins it would be great if authors could test whether it also works with MMLV Gag VLPs.

6. Although the technical quality of the manuscript is excellent, one remains with an important unanswered question. How well does DIRECTED perform in primary cells? Primary T cells are a good cell-type candidate that authors could test with DIRECTED, either targeting CD3 or even better, CD4 or CD8 to show the specificity. Testing DIRECTED in vivo would also be a great addition but I know those experiments can be long and will not necessarily improve the message of the manuscript (maybe targeting the liver would be an easy way to show in vivo feasibility).

Minor points:

Line 274-275 : Authors mention that they established Cre delivery to recipient cells without DIRECTED and point to Supplementary Figures 4a and b to sustain their claim. However, Supplementary Figure 4a corresponds to western-blot analysis of Cre VLPs using anti-Cre and VSV-G antibodies and Supplementary Figure 4b corresponds to in vitro Cre reactions. Did authors mean that they established « Cre loading within VLPs » instead of « Cre delivery to recipient cells » ?

Reviewer #3:

Remarks to the Author:

This manuscript by Streibinger et al. reports convincing in vitro results of the proof-of-concept for the development of a modular platform they called DIRECTED (DELIVERY to Intended REcipient Cells Through Envelope Design) for cell type-specific delivery of cargos including transgenes by lentiviral vectors, but also proteins and nucleic acids by empty virus-like particles (eVLP). The rationale of the DIRECTED platform is to dissociate virus-cell fusion and cell-targeting by using modified forms of viral envelopes, mainly the G protein of VSG but also other viral envelopes, still efficient for cell-fusion but defective in cell-receptor recognition, in association with antibody targeting for cell-specific delivery of the cargo. While this concept and design for cell-type specific targeting seems to work efficiently for cargo delivering in vitro, it is difficult to imagine how this system could work and be developed for in vivo applications, as mentioned by the authors in Discussion section. More specifically, all the results reported in the present manuscript using this DIRECTED delivery system for cell-type specific targeting have been performed in rather easy to transduce human transformed cell-lines, such as embryonic kidney HEK293T cells and modified Jurkat lymphoid cells, but analysis for specific delivering at least in human primary cells, or even better ex vivo in complex tissues or organoids, is needed to properly evaluate the strengths and weaknesses of this new delivery system compared to the existing delivery systems.

As mentioned, the major point to significantly strengthen the intrinsic and general impact of the present manuscript is to evaluate the effectiveness and specificity of the DIRECTED system for

cargo delivering in human adherent and/or non-adherent primary cells such as primary lymphoid blood cells or endothelial or epithelial primary cells for comparison with other delivery systems. For example, the use of PBMCs (peripheral blood mononuclear cells) could be a very valuable simple model for analysis of cargo delivering in the different blood immune cell populations for cell-type specific targeting using specific antibodies of the different cell-types (i.e., subsets of B and T lymphocytes, monocytes, NK cells).

Specific points:

1) In Figure 1 regarding the development of the DIRECTED system using the protein AG and anti-HLA-A2 antibody for transgene delivering by lentiviral vectors in HEK293T cells, the authors should include experiments using the HEK293T B2M KO cell-line, described in the supplementary Figure 1C as a nice and robust control of specificity.

2) Figure 2: The authors should explain, or at least comment, why the level of transduction reported with the 3 different systems and reported in panel A is around a maximum of 20%, whereas around 85% of this modified Jurkat cells express HA (see supplementary Figure 2a)? The authors could also comment why the levels of transduction obtained in HEK293T cells is much higher than those obtained in Jurkat cells.

3) As indicated for the Figure 1 data, the authors should include to the data reported in panel C (using the Cocal virus G envelope) additional experiments using HEK293T B2M KO cells as control.

4) Finally, I suggest the authors to totally rewrite the Abstract that is too much general in its present form to describe more specifically the rationale and the experimental approaches used for the development of the different DIRECTED platforms described, as well as the main results obtained using these different platforms. In addition, they could give more details in the figure legends for a better and easier understanding of the results shown.

Reviewer #1 (Remarks to the Author):

This manuscript shows an elegant approach to redirect enveloped viral vectors, in particular pseudotyped lentiviral vectors towards target cells, making use of 3 different approaches to display antibodies or antibody fragments on the surface of the viral envelopes. The strength of this approach is the ability to use commercial antibodies for redirecting and the option for covalent attachment as is the case for the SNAP-tag approach. In addition, the authors show that this REDIRECT approach can work with different fusogens, including VSV-G and mutant versions that lack LDL-R binding.

The ability to redirect viral vectors for cell-type specific transduction is a long held desire and a need for direct in vivo modification of cells (e.g. generation of CAR T cells directly in vivo). Several approaches have been evaluated, including envelope engineering, pseudotyping as well as redirecting lentiviral vectors using bispecific antibodies.

Even though this manuscript nicely shows proof-of-concept for target-cell specific transduction, the redirection of viral vectors is not novel. Several groups have recently shown that this is feasible by using bispecific antibodies (<https://doi.org/10.1128/mBio.02990-19>; <https://doi.org/10.1136/jitc-2021-002737>; <https://doi.org/10.3390/v14102157>). The use of bispecific antibodies has as main advantage that no engineering of the viral envelope is needed, which favours high titres. It would be interesting to learn how these two approaches relate to each other and I therefore suggest a direct comparison.

We appreciate the Reviewer's assessment of our work and have now cited the references they suggested in the introduction. While other strategies such as bispecific antibodies have shown that viral vector redirection is feasible without viral envelope engineering, the DIRECTED approach uniquely uses unmodified viral fusion proteins in combination with targeting molecules to expand viral tropism by enabling the use of an additional receptor specified by the targeting molecule, as demonstrated in Figure 1b.

Regarding the comparison with bispecific antibodies, we were unable to find the sequences for the antibodies mentioned in the references or publicly available databases, so we were unable to perform a direct comparison. However, the presence of the ZZ domain in every copy of the envelope protein (as in m168) mimics aspects of the bispecific antibody strategy. Therefore, we compared the m168 pseudotyped particles with DIRECTED (VSVG K47Q, R354Q and proteinAG-TM) pseudotyped particles, and found that both produce similar levels of viral titers (revised Supplementary Figure 4a). Furthermore, they perform with equal efficiency on HEK293FT cells engineered to express a Surface-Spot synthetic receptor. While the VSVG K47Q, R354Q mutant retains a higher level of residual activity on HEK293FT cells, the use of protein AG has the advantage of efficiently binding antibodies from a wider range of species compared to the ZZ domain of protein A (see <https://www.sigmaaldrich.com/US/en/technical-documents/technical-article/protein-biology/protein-pulldown/protein-a-g-binding>). Specifically, protein A has low efficiency for binding to IgGs from rat, which are commonly used for the generation of mouse protein targeting antibodies. Additionally, the modularity of DIRECTED allows for the exchange of the protein AG moiety for other molecules, such as the presented SNAP strategy, further expanding the set of biomolecules that can be immobilized on the surface of virions. Our own experiments have shown that the ZZ domain of protein A in m168 cannot be replaced by a SNAP tag to enable a similar strategy with this envelope (data not shown).

Furthermore, all experiments presented were performed in vitro on cell types with high expression levels of a target receptor or with artificial receptors introduced by lentiviral transduction. This is rather artificial and what is missing in my opinion are experiments to demonstrate that this REDIRECT approach would also work in a more complex environment, for example cell-type specific transduction of T cells or NK cells in whole blood. This would really demonstrate the value of the REDIRECT approach. As the authors

already point out, some of the approaches might not work in vivo due to e.g. inactivation of VSV-G by human serum components, competition of IgG for protA etc. etc. It would be very informative to learn which of the combinations of fusogens and antibody attachment methods would indeed give robust in vivo or ex vivo transduction.

We concur with the Reviewer's suggestion that investigating the efficacy of various combinations of DIRECTED components for specific applications would be a compelling avenue for future research. Our primary focus in this study was to investigate the different approaches that facilitate the attachment of antibodies to the viral surface. We have extended our initial tests toward this future goal, however, demonstrating that DIRECTED can deliver payloads (lentiviral, and Cas9-RNPs) to primary human T cells using α CD3 targeting (Figures 4d,e).

Minor comments:

Statistics are missing for all figures. With only mean values and the individual data points, it is difficult to estimate if some values are significantly different.

We added statistics where appropriate.

"We next sought to block the intrinsic VSV-G tropism through addition of a competitor molecule (Supplementary Figure 1f)" S1F shows HLA-A2 targeted data, not inhibition with a competitor molecule recombinant diCR2 protein.

We apologize for the confusion. Supplementary Figure 1f (which is now Supplementary Figure 1h in the revised manuscript) shows the performance of wildtype VSV-G+pAG pseudotyped particles in the presence of a soluble competitor protein (diCR2) in the absence or presence of an α HLA-A2-targeting antibody. We added an inset to the panel to make this more obvious.

Unclear if transduction of cells was performed in presence or absence of FBS. Please add info to M&M

We apologize for the confusion. This information has been more explicitly described in the Materials and Methods.

Efficiency of transduction with the SNAP-REDIRECTED approach (Fig 2C) seems to be very much dependent on type of antibody. Did the authors test different ratios of the benzylguanine label to protein?

We agree with the Reviewer's observation. We tested a 10-fold, 20-fold, 40-fold, and 80-fold excess of BG-GLA-NHS to antibody, and observed that at the 80-fold ratio, there was a risk of protein precipitation and loss, which was less obvious with a 40-fold excess. Nevertheless, the use of NHS-esters for labelling could affect antibody solubility and function, especially if Lysine residues occur in the complementary determining regions. As the sequences of commercial antibodies are typically unknown, different ratios should be tested for new antibodies. As an example, we present the performance of SNAP-DIRECTED lentiviral vectors with the α Spot antibody labelled at the above-mentioned ratios in Supplementary Figure 3a, which demonstrates that this specific antibody is inhibited by higher amounts of BG-GLA-NHS. We have added a discussion of this limitation in more detail in the Discussion section of the paper.

Reviewer #2 (Remarks to the Author):

In this manuscript, Streibinger D et al., describe a modular viral envelope design consisting on separate fusion and targeting moieties that achieve cell specific transgene, protein and RNP delivery. The methodology, dubbed DIRECTED, relies on the co-expression of a viral fusogen (mutated or not to block its capacity to interact with a specific cellular receptor) and a transmembrane protein which is engineered to bind antibodies and allow users to define the tropism of the produced viral particles.

Using DIRECTED with wild-type VSV-G as a fusogen and a targeting moiety consisting of the VSV-G transmembrane domain fused to pAG, authors produced lentiviral vector particles that were then programmed with an HLA-A2-targeting antibody. Produced particles were able to transduce HEK293FT cells in a partially antibody-dependent manner. Different mutants of VSV-G were then generated to abolish its binding capacity and rely exclusively on the pAG protein to specify targeting. This approach significantly reduced off-target binding in the absence of a bound antibody, while still allowing high lentiviral titers and efficient and specific transduction of target cells in the presence of a targeting antibody. Authors then engineered additional targeting moieties relying on a single chain variable fragment (scFv) or SNAP-tag fused to the VSV-G transmembrane domain which, similarly to pAG, also led to cell-specific targeting, each with variable efficiency depending on the nature of the targeted surface receptor.

Having validated DIRECTED, authors then performed a sequence-based homology search for other fusogens related to VSV-G, that led to the identification of several fusogens from the Rhabdoviridae family (one, Cocal virus G protein, being functional to replace VSV-G). The search for alternative fusogens was then expanded to a library of fusogens from different viral families, which further identified several functional fusogens compatible with the DIRECTED approach. Among these, GP64 from baculovirus performed efficiently, therefore expanding the list of fusogens compatible with DIRECTED that could be specifically useful for in vivo studies where VSV-G is known to be rapidly inactivated by serum proteins.

Finally, authors show that DIRECTED is compatible with retroviral Gag proteins in addition to the lentiviral Gag protein used throughout the previous figures. More particularly, they validated DIRECTED with Cre- and Cas9-VLPs for direct protein and RNP delivery.

Overall the manuscript is of excellent technical quality and tackles one of the last obstacles limiting the use of viral vectors in vivo and in cell types that are difficult to target with unmodified viral envelopes. Although the technical approach is not entirely novel at conceptual level (see point 1 below), the authors go beyond what has been already published regarding the optimization of their platform (although experiments with primary cells or in vivo are missing). Furthermore, the identification of new compatible fusogens (other than the mutant VSV-G) will be of great interest for scientists working in a wide-range of research fields.

We thank the Reviewer for their positive assessment of our work.

Below you will find some major and minor points that, in my opinion, should be addressed by authors in order to improve the manuscript.

1. Line 78-80 : In the introduction, authors mention a recent manuscript (Dobson CS et al., 2022) that describes a VSV-G variant that has lost affinity for LDL-R and which can be used to identify antigen-specific T cells. However, the method described in that manuscript is not restricted to the identification of antigen-specific T cells, but was also used by authors to mediate cell type-specific lentiviral delivery of transgenes in a conceptually similar modular platform as DIRECTED, combining a mutated VSV-G (bearing mutations at the same residues as those selected for the optimal DIRECTED protocol, K47 and R354) with a broad range of targeting moieties. This information should be clearly acknowledged in the manuscript text, for example in lines 114-115 when authors mention the idea of a modular platform.

Moreover, there was a second manuscript published in the same issue as Dobson CS et al., that should be cited as well in line 78-80 (<https://doi.org/10.1038/s41592-022-01437-y>).

Finally, the chimeric Sindbis virus envelope m168 that can be conjugated with an antibody similarly to DIRECTED was successfully validated in vivo and shown to specifically target melanoma cells in mice (<https://www.nature.com/articles/nm1192>). This manuscript should also be cited so that readers are aware of the pre-existing literature related to DIRECTED.

We thank the Reviewer for drawing this to our attention, and we have added the mentioned citations and information to the introduction.

2. Authors indicate the MOI (calculated as VGs/cell) used for DIRECTED experiments starting from Figure 1d. However, it will be informative to compare the MOIs obtained with WT VSV-G to those obtained with DIRECTED in order to know the relative efficiency of each setup. It is also important for authors to show cell viability upon transduction with VSV-G pseudotyped particles and when using DIRECTED (one antibody type will be enough) both for HEK293FT and Jurkat cells.

We thank the Reviewer for their comment. We added a direct comparison of pAG DIRECTED lentiviral vectors using GP64, COCVmut, and VSV-Gdm with wildtype VSV-G at the same MOI in the presence or absence of an α HLA-A2-targeting antibody in Supplementary Figure 4g. This comparison revealed that all pseudotypes perform comparably to wildtype VSV-G upon targeting of HLA-A2 on HEK293FT cells. Moreover, we also show the absolute production efficiency for each of these combinations compared to wildtype VSV-G as determined by RT-qPCR (expressed as VGs/ μ l).

To evaluate the cell viability upon transduction of cells with DIRECTED lentiviral vectors or wildtype VSV-G, we infected HEK293FT + surface-Spot cells for 24h with VSV-Gdm+pAG+ α Spot or wildtype VSV-G at the same MOI (300 VGs/cell), which revealed no significant differences in cell number or the percentage of Annexin V positive cells four days after transduction (Supplementary Figure 3b). As a control we infected cells with very high doses of wildtype VSV-G (MOI=10.000VGs/cell) which showed a significant reduction in overall cell numbers and an increase in the fraction of Annexin V positive cells. This MOI is higher than the MOIs we used in all other experiments with HEK293FT cells presented throughout the manuscript.

We also performed similar experiments with Jurkat E6 cells using a VSV-Gdm+SNAP DIRECTED lentiviral vector that was either non-functionalized or clicked with α CD3 antibody (Supplementary Figure 3c). We transduced these cells using spinoculation and Polybrene (which we found to increase transduction efficiency between 2-10-fold – Supplementary Figure 2a) at an MOI of 50, revealing no significant changes in cell numbers, but an increase in the fraction of Annexin V positive cells with the α CD3-DIRECTED particles. Additionally, we used a wildtype VSV-G at an MOI of 400 as an extreme case, which showed a significant decrease in cell numbers upon transduction and increase in the fraction of Annexin V positive cells.

3. Most surface receptors targeted correspond to abundant proteins. It would be interesting for future users to know how well DIRECTED performs when surface receptors are less abundant. Could authors sort Surface-HA cells to obtain populations displaying different HA expression levels and test how well DIRECTED performs at each expression level of the surface receptor?

We thank the Reviewer for this suggestion. We have now performed experiments using our Jurkat Surface-HA cell line, which we sorted into 4 bins of different expression levels. We then measured the mean fluorescence intensity in each of these bins upon staining with an α HA-PB450 antibody, allowing us to assign relative expression levels (Supplementary Figure 2f,i). Using these cells, we tested their transduction

efficiency with our three strategies, finding that the genetically encoded α HA-scFv approach resulted in high transduction efficiency even at low receptor levels (Figure 2a). While all approaches displayed a similar transduction efficiency for the high expressing bin, the pAG and SNAP strategies performed worse at low surface receptor levels. We suspected this may be due to the presence of excess antibody and removed these from SNAP clicked particles by ultracentrifugation through a 20% sucrose cushion. Using this purified α HA-SNAP-DIRECTED preparation, we observed transduction efficiencies that were similar to the scFv strategy (Figure 2a and Supplementary Figure 2h,i). This suggests that the pAG strategy is less robust when targeting lowly expressed receptors and may require testing of different antibody amounts, whereas the scFv and SNAP (if purified from excess antibody) strategies perform robustly even when targeting less abundant receptors.

We also added experiments targeting CD3 and CD5 on Jurkat cells, both of which are endogenously expressed and show a similar expression profile (Figure 4c). Using pAG and SNAP DIRECTED eVLPs, we showed that while pAG DIRECTED eVLPs in the presence of CD5 plateau around 50% knockout efficiency, α CD5-BG SNAP DIRECTED particles perform similar to α CD3-BG SNAP DIRECTED particles. Together with the data presented in Supplementary Figure 3e,f, these results suggest that the SNAP strategy is more robust when targeting a broad range of surface receptors. This may be explained by the different dynamics between the pAG and SNAP strategies: While the pAG strategy requires two independent binding events (binding of the antibody to the receptor and binding of the pAG to the antibody, both with different affinities), the SNAP strategy only requires one binding event, which is solely determined by the affinity of the immobilized antibody for the receptor.

4. The transmembrane domain of VSV-G used in the targeting module contains residues that are critical for the monomeric to trimeric conformation and subsequent hemi-fusion process (<https://www.nature.com/articles/s41598-018-28868-y>, <https://www.pnas.org/doi/10.1073/pnas.95.7.3425>, <https://www.pnas.org/doi/10.1073/pnas.1618883114>). Could this perturb core VSV-G trimer formation at the hemi-fusion step and explain the decrease in transduction efficiency observed in Figure 1e when adding large amounts of antibodies.

We thank the Reviewer for their hypothesis. We initially also considered the possibility that the VSV-G transmembrane domain on our targeting molecule could interfere with VSV-G trimer formation. This was one of the main motivations for us to test different ratios of VSV-G:pAG (Figure 1b). We hypothesized that if trimer formation is impaired, we should see a significant reduction in the overall transduction potential of particles that are decorated with both wildtype VSV-G and the protein AG variant. We see an increase in transduction of cells with a low percentage of protein AG in the presence of the α HLA-A2 antibody, surpassing the levels observed with wildtype VSV-G only, suggesting that trimer formation is intact. Moreover, statistical analysis of the data in Figure 1e revealed that there is no significant difference even at high antibody concentrations.

5. Figure 4: Testing DIRECTED with MMLV Gag VLPs was done with the mutant VSV-G fusogen only. Since GP64 is a good candidate for in vivo experiments in order to circumvent VSV-G inactivation by serum proteins it would be great if authors could test whether it also works with MMLV Gag VLPs.

We thank the Reviewer for this suggestion. We have now added experiments to evaluate DIRECTED with MMLV gag VLPs using GP64 and COCVmut as additional fusion proteins. Both proteins were indeed compatible with MMLV gag VLPs and allowed the efficient delivery of B2M-targeting Cas9-RNPs into Jurkat cells (Figure 4b).

6. Although the technical quality of the manuscript is excellent, one remains with an important unanswered question. How well does DIRECTED perform in primary cells? Primary T cells are a good cell-type candidate that authors could test with DIRECTED, either targeting CD3 or even better, CD4 or CD8 to

show the specificity. Testing DIRECTED in vivo would also be a great addition but I know those experiments can be long and will not necessarily improve the message of the manuscript (maybe targeting the liver would be an easy way to show in vivo feasibility).

We thank the Reviewer for raising this critical point. We have now performed experiments on primary human T cells, with both α CD3 SNAP DIRECTED lentiviral vectors and α CD3 SNAP DIRECTED eVLPs (Figure 4d,e). We also evaluated targeting of CD4 and CD8, however neither of these targets showed functional delivery in our hands.

Minor points:

Line 274-275 : Authors mention that they established Cre delivery to recipient cells without DIRECTED and point to Supplementary Figures 4a and b to sustain their claim. However, Supplementary Figure 4a corresponds to western-blot analysis of Cre VLPs using anti-Cre and VSV -G antibodies and Supplementary Figure 4b corresponds to in vitro Cre reactions. Did authors mean that they established « Cre loading within VLPs » instead of « Cre delivery to recipient cells » ?

We thank the Reviewer for spotting this and we have corrected the text accordingly.

Reviewer #3 (Remarks to the Author):

This manuscript by Strebinger et al. reports convincing in vitro results of the proof-of-concept for the development of a modular platform they called DIRECTED (DELIVERY to Intended REcipient Cells Through Envelope Design) for cell type-specific delivery of cargos including transgenes by lentiviral vectors, but also proteins and nucleic acids by empty virus-like particles (eVLP). The rationale of the DIRECTED platform is to dissociate virus-cell fusion and cell-targeting by using modified forms of viral envelopes, mainly the G protein of VSG but also other viral envelopes, still efficient for cell-fusion but defective in cell-receptor recognition, in association with antibody targeting for cell-specific delivery of the cargo. While this concept and design for cell-type specific targeting seems to work efficiently for cargo delivering in vitro, it is difficult to imagine how this system could work and be developed for in vivo applications, as mentioned by the authors in Discussion section. More specifically, all the results reported in the present manuscript using this DIRECTED delivery system for cell-type specific targeting have been performed in rather easy to transduce human transformed cell-lines, such as embryonic kidney HEK293T cells and modified Jurkat lymphoid cells, but analysis for specific delivering at least in human primary cells, or even better ex vivo in complex tissues or organoids, is needed to properly evaluate the strengths and weaknesses of this new delivery system compared to the existing delivery systems.

As mentioned, the major point to significantly strengthen the intrinsic and general impact of the present manuscript is to evaluate the effectiveness and specificity of the DIRECTED system for cargo delivering in human adherent and/or non-adherent primary cells such as primary lymphoid blood cells or endothelial or epithelial primary cells for comparison with other delivery systems. For example, the use of PBMCs (peripheral blood mononuclear cells) could be a very valuable simple model for analysis of cargo delivering in the different blood immune cell populations for cell-type specific targeting using specific antibodies of the different cell-types (i.e., subsets of B and T lymphocytes, monocytes, NK cells).

We thank the Reviewer for raising this critical point. We have now performed experiments on primary human T cells, with both α CD3 SNAP DIRECTED lentiviral vectors and α CD3 SNAP DIRECTED eVLPs (Figure 4d,e). These results strengthen the data presented in the original manuscript in Figure 1f which show that the DIRECTED system can achieve specific delivery to a desired cell type within a mixed population.

Specific points:

1) In Figure 1 regarding the development of the DIRECTED system using the protein AG and anti-HLA-A2 antibody for transgene delivering by lentiviral vectors in HEK293T cells, the authors should include experiments using the HEK293T B2M KO cell-line, described in the supplementary Figure 1C as a nice and robust control of specificity.

We have added control experiments with B2M knockout cells (Supplementary Figure 1d and Figure 1d).

2) Figure 2: The authors should explain, or at least comment, why the level of transduction reported with the 3 different systems and reported in panel A is around a maximum of 20%, whereas around 85% of this modified Jurkat cells express HA (see supplementary Figure 2a)? The authors could also comment why the levels of transduction obtained in HEK293T cells is much higher than those obtained in Jurkat cells.

We thank the Reviewer for this comment. We think that the truncated PDGFRb transmembrane domain, which we used to engineer the synthetic surface-HA receptor, doesn't allow efficient turnover/internalization in Jurkat cells. We hypothesize this may be due to the absence of endogenous PDGFRb expression in Jurkat cells, and thus missing regulators for this protein. The DIRECTED system relies on pH-dependent fusogens (either with natural tropism or blinded for intrinsic receptor binding), which require

a low pH for the fusogen to become activated and perform membrane fusion. Therefore, the mere binding of a receptor is not enough to enable successful delivery, but rather the receptor-virus complex needs to be internalized, where upon encountering of low pH in the endosomal compartment the fusogen will be triggered. If the internalization or the turnover of the receptor is inefficient, this will result in reduced delivery potential. We have included this hypothesis in the revised version of the manuscript.

Jurkat cells infect at lower efficiency in our hands, as shown in Supplementary Figure 4d,e. However, to boost transduction in Jurkat cells, we evaluated the performance of transduction enhancers (Polybrene and vectofusin) and spinoculation (Supplementary Figure 2a). We observed that 8 $\mu\text{g/ml}$ Polybrene resulted in a strong increase in the fluorescence signal of transduced cells for all tested envelopes, which was further increased by using spinoculation.

3) As indicated for the Figure 1 data, the authors should include to the data reported in panel C (using the Cocal virus G envelope) additional experiments using HEK293T B2M KO cells as control.

We have added control experiments with B2M knockout cells (Figure 3C and 3F).

4) Finally, I suggest the authors to totally rewrite the Abstract that is too much general in its present form to describe more specifically the rationale and the experimental approaches used for the development of the different DIRECTED platforms described, as well as the main results obtained using these different platforms. In addition, they could give more details in the figure legends for a better and easier understanding of the results shown.

We have revised the abstract to be more specific and included more details in the figure legends to better guide the reader.

Reviewers' Comments:

Reviewer #1:

Remarks to the Author:

I thank the authors for their careful consideration of all the points raised by the reviewers. The revised manuscript has certainly improved, but is in my opinion still lacking a very important set of data, which was requested by all 3 reviewers independently, namely showing proof of concept that the REDIRECT strategy will indeed work in vivo, or at least in conditions close to the in vivo situation. Even though the authors have added new data to show targeting to T cells, these are based on ex vivo transduction of isolated T cells, for which many other technologies can be applied (e.g. electroporation, lentiviral transduction, lipofection etc.). Therefore the added benefit of those experiments are limited. The novelty and real impact would be in direct in vivo targeting of e.g. circulating T cells. It is my strong opinion that these data should be included for publication, irrespective of the outcome of such in vivo targeting experiments. A simple experiment where PBMCs isolated from human donor blood is incubated with the REDIRECT viral vectors, after which each subpopulation of cells is checked by flow cytometry for transduction will in my opinion already be sufficient if setting up an in vivo experiment would take too much time/effort.

Reviewer #2:

Remarks to the Author:

The authors have satisfactorily addressed all my remarks/suggestions. Results obtained using cells expressing different levels of the surface HA receptor revealed some differences among the pAG, scFv and SNAP strategies that will be important for potential users when deciding which strategy to use. Similarly, the improvement obtained with SNAP in targeting lowly abundant receptors when removing antibodies in excess will also be helpful for users. Finally, I also appreciate the new discussion that describes the advantages but also some of the limitations of DIRECTED.

The manuscript is, in my opinion, suited for publication.

Reviewer #3:

Remarks to the Author:

As requested in the previous review, the authors now included some additional controls and changes in the revised version of the manuscript, but the major concern of my previous review was to evaluate the efficiency and specificity of the DIRECTED system for cargo delivering in human primary cells in a more complex system containing distinct cell populations such as PBMCs, and comparison with other more classic delivery systems. Regarding this major point, the authors now show additional data, reported in Figure 4d and 4e (and in Supplementary Figure 5i and 5j), for cargo delivering using purified blood T cells and not PBMCs. Moreover, I am wondering about the low transduction efficiency reached (around 3%), using so high MOI (i.e., 1500)? In addition, there is no comparison with other delivering systems in the new data presented.

We revised our manuscript and performed additional experiments to address the reviewer comments. Specifically, we included experiments showing the successful targeting of resting T cells and B cells in a complex mixture of PBMCs (Figure 5b,c and Supplementary Figure 6). Our experiments also showed that we could functionalize DIRECTED particles with multiple ligands (Figure 5b,c and Supplementary Figure 6e). Furthermore, we tested DIRECTED viral particles for T cell targeting in whole blood and showed that DIRECTED enables successful transduction of primary human T cells in these conditions (Figure 5e Supplementary Figure 7). Finally, we evaluated the in vivo performance of DIRECTED CreVLPs to target circulating T cells using an α CD5 antibody. These experiments revealed changes in splenic cell type composition, and an increase in F4/80+ macrophages in all conditions, regardless of targeting moiety. Many of these macrophages were positive for tdTomato. While this off-target transduction is a major limitation of DIRECTED, it also represents a major problem for other delivery modalities such as LNPs.

Reviewer #1 (Remarks to the Author):

I thank the authors for their careful consideration of all the points raised by the reviewers. The revised manuscript has certainly improved, but is in my opinion still lacking a very important set of data, which was requested by all 3 reviewers independently, namely showing proof of concept that the REDIRECT strategy will indeed work in vivo, or at least in conditions close to the in vivo situation. Even though the authors have added new data to show targeting to T cells, these are based on ex vivo transduction of isolated T cells, for which many other technologies can be applied (e.g. electroporation, lentiviral transduction, lipofection etc.). Therefore the added benefit of those experiments are limited. The novelty and real impact would be in direct in vivo targeting of e.g. circulating T cells. It is my strong opinion that these data should be included for publication, irrespective of the outcome of such in vivo targeting experiments.

We have sought to address this comment in our revised study by performing experiments using CreVLPs for the in vivo targeting of circulating T cells using α CD5 targeting in immunocompetent BL6 Cre reporter animals. We chose CreVLPs as we expect their delivery to be subject to less host factor restriction than lentiviral vectors. We found that wild-type VSV-G pseudotyped particles resulted in the highest overall frequency of tdTomato positive cells in the spleen of treated animals (Supplementary Figure 8b). Moreover, the cell type composition in VSV-G treated animals changed significantly with an increase in CD3+ cells and a reduction in CD20+ cells (Supplementary Figure 8a), which may indicate an inflammatory response. Comparing DIRECTED SNAP Cre-VLPs in the absence of an antibody or in the presence of a CD5 antibody showed no significant changes in cell type composition and more importantly similar transduction efficiency for both variants. We did observe a trend for higher transduction efficiency of CD3+ cells for the CD5-DIRECTED particles, but it was not statistically significant (Supplementary Figure 8b). We also observed reduced transduction of monocytes and B cells compared to VSV-G for CreVLPs in the absence and presence of α CD5. Furthermore, we observed splenomegaly in the treated animals, and could detect an increase of F4/80+ cells in all treated animals (Supplementary Figure 8d, e). We think that this may result from the transduction of circulating monocytes in peripheral blood, which subsequently differentiate into macrophages and migrate to the spleen. The transduction efficiencies we could detect for CD3+ cells in using CD5-DIRECTED CreVLPs does however fall within a range that could allow for the generation of CAR-T cells in vivo in a cancer model, in which the transduced cells would further amplify upon antigen encounter. We have included this data in the manuscript, in line with the Reviewer's suggestion.

A simple experiment where PBMCs isolated from human donor blood is incubated with the REDIRECT viral vectors, after which each subpopulation of cells is checked by flow cytometry for transduction will in my opinion already be sufficient if setting up an in vivo experiment would take too much time/effort.

In addition to the above-mentioned in vivo experiment, we have now explored the functionality of SNAP-DIRECTED lentiviral vectors on PBMCs and in whole blood. Due to a lack of culture conditions that allow for the successful long-term maintenance and expansion of the diverse cell types in peripheral blood, we focused our analyses mainly on the T cell compartment. We showed that the combination of wild-type VSV-G with SNAP and α CD3-BG increases transduction of naïve human T cells in a complex mixture of cell types (Supplementary Figure 6a). Moreover, using the VSV-G mutant variant (VSV-Gdm), we found that SNAP-DIRECTED lentiviral particles functionalized with α CD3 or an antibody mix targeting multiple receptors (α CD3, α CD28, α CD4) on human T cells successfully transduce naïve human T cells at a higher efficiency than wild-type VSV-G, while there is no transduction in the presence of B cell targeting ligands (α CD19, MegaCD40L) or in the absence of an antibody (Figure 5b). We also found that the presence of the CD4 antibody biases DIRECTED particles that were functionalized with the T cell targeting antibody mix towards CD4+ cells (Supplementary Figure 6e). Furthermore, we showed that B cell targeting DIRECTED lentiviral vectors result in the successful transduction of B cells, with no transduction of T cells (Figure 5c, Supplementary Figure 6g). One caveat to these experiments, however, is the low number of surviving B cells, suggesting that an optimization of the culture conditions would be necessary to expand these cells.

We then explored the performance of T cell targeting SNAP DIRECTED lentiviral particles (again using α CD3 or the antibody mix α CD3, α CD28, α CD4) on whole blood. Given the complex environment and the limited availability of essential survival factors, we had to limit the time the lentiviral particles could interact with the cells to 6 hours before

we lysed red blood cells and cultured the remaining cells in T cell media (RPMI + 10% FBS + rhIL-2 + rhIL-15 + rhIL-7 + α CD3, α CD28 T cell activator beads) to ensure their survival. The additional host restriction factors present in whole blood (i.e., neutralizing antibodies, a functional complement system, and phagocytotic cells), the high cell density, and the limited exposure time of cells to the virus as compared to other in vitro experiments could result in lower transduction efficiency than observed in other experiments. Analysis of the cells showed that the T cell mix resulted in the transduction of ~1.5% of CD4+ T cells, which is ~3-fold more than wild-type VSV-G (Figure 5e). Moreover, the transduction efficiency in whole blood was in the same range as previously reported values for VSV-G and engineered CD3 targeting viruses in whole blood (<https://ashpublications.org/bloodadvances/article/4/22/5702/474209/Combining-T-cell-specific-activation-and-in-vivo>). The transduction level we achieved is also of biological relevance because the range of specific T cell clones in viral infection can reach up to 2% (<https://journals.plos.org/plosone/article?id=10.1371/journal.pone.0000649#pone.0000649-Seth1>), which is already after these clones expanded. If used in an in vivo CAR-T setting, the 1.5% of cells that we could transduce may benefit from selective expansion. Finally, the possibility of functionalizing SNAP DIRECTED particles with multiple targeting ligands and/or agonistic molecules will also allow for coupling viral transduction to the delivery of cellular signals to explore the role of these signals in biological processes. We have now included this data in the manuscript.

Reviewer #2 (Remarks to the Author):

The authors have satisfactorily addressed all my remarks/suggestions. Results obtained using cells expressing different levels of the surface HA receptor revealed some differences among the pAG, scFv and SNAP strategies that will be important for potential users when deciding which strategy to use. Similarly, the improvement obtained with SNAP in targeting lowly abundant receptors when removing antibodies in excess will also be helpful for users. Finally, I also appreciate the new discussion that describes the advantages but also some of the limitations of DIRECTED.

The manuscript is, in my opinion, suited for publication.

We thank the reviewer for their positive assessment of our work.

Reviewer #3 (Remarks to the Author):

As requested in the previous review, the authors now included some additional controls and changes in the revised version of the manuscript, but the major concern of my previous review was to evaluate the efficiency and specificity of the DIRECTED system for cargo delivering in human primary cells in a more complex system containing distinct cell populations such as PBMCs, and comparison with other more classic delivery systems. Regarding this major point, the authors now show additional data, reported in Figure 4d and 4e (and in Supplementary Figure 5i and 5j), for cargo delivering using purified blood T cells and not PBMCs.

Please see our response to Reviewer 1 regarding targeting in whole blood and in vivo.

Moreover, I am wondering about the low transduction efficiency reached (around 3%), using so high MOI (i.e., 1500)? In addition, there is no comparison with other delivering systems in the new data presented.

We thank the Reviewer for this question. While we do not fully understand why α CD3 DIRECTED lentiviral particles do not perform well on activated primary human T cells, this is in line with other work that uses scFvs to target human CD3 on activated primary human T cells (<https://www.biorxiv.org/content/10.1101/2022.08.24.505004v2.full.pdf>). We also want to highlight that the indicated MOI presents the number of viral genomes (VGs) per cell as determined by RT-qPCR after benzonase treatment. As this assay does not measure functional viral vectors, it tends to be higher than functional titers. Due to the absence of reliable tropism of VSV-Gdm, we use this measure throughout the manuscript. To give an estimate of these numbers, Jurkat cells need an MOI of 500 to achieve ~30% transduction with pAG-DIRECTED particles targeting CD3 (Figure 1e) and Kasumi-1 cells tend to transduce less efficiently than most other cell types and require an MOI of ~1000 to achieve 20% transduction rate using wild-type VSV-G (Supplementary Figure 3h). We included m168 (blinded sindbis virus envelope with inserted protein A domain) in our previously reported T cell data, we did not include a VSV-G pseudotyped control. We found that m168 did not allow successful transduction of T cells in the presence of an α CD3 antibody. However, to avoid confusion we removed these data from the manuscript and replaced it with our PBMC and whole blood experiments (Figure 5, Supplementary Figures 6 and 7).

For all experiments to evaluate the targeting of T cells and B cells in PBMCs, as well as T cell targeting in whole blood and the in vivo experiment include wild-type VSV-G as a comparison to a more traditional delivery system. As VSV-G is the most commonly used envelope for the pseudotyping of lentiviral vectors, we reasoned that this would be the best comparison for our system. Furthermore, the fact that LNPs do not integrate, would have made direct comparisons more difficult, as our readouts were performed on day 6 and day 14 post treatment. These experiments showed that our DIRECTED lentiviral particles outperformed wild-type VSV-G on resting T cells or B cells in PBMCs (Figure 5b, c)

and performed at equal efficiency on T cells in whole blood (Figure 5e). Moreover, we also use the VSV-G mutant version (VSV-Gdm) as a negative control to show the dependence of the system on the presence of the targeting molecules. This showed that while wild-type VSV-G allows the transduction of T cells in PBMCs, the VSV-G mutant (VSV-Gdm) is incapable of transducing T cells and depends on antibodies or ligands that target specific cell types to achieve successful transgene delivery (Figure 5b and Supplementary Figure 6e).

Reviewers' Comments:

Reviewer #1:

Remarks to the Author:

I am pleased to see the new results on direct transfection of T- and B- cells directly from whole blood. With this addition, the manuscript now gives a complete and balanced overview of the capabilities of the DIRECTED vector technology. I recommend publication of the manuscript in its present form.

Enrico Mastrobattista

Reviewer #3:

Remarks to the Author:

Even if the authors now include, especially in the new figure 5, additional data using the DIRECTED system to target T or B cell population, I am still not convinced that this system brings significant improvement compared to the available transduction system, leading to low level of transduction using so high MOI.

Reviewer #1 (Remarks to the Author):

I am pleased to see the new results on direct transfection of T- and B- cells directly from whole blood. With this addition, the manuscript now gives a complete and balanced overview of the capabilities of the DIRECTED vector technology. I recommend publication of the manuscript in its present form.

Enrico Mastrobattista

We thank the reviewer for his positive assessment of our work, his suggestions, and his time to review our manuscript.

Reviewer #3 (Remarks to the Author):

Even if the authors now include, especially in the new figure 5, additional data using the DIRECTED system to target T or B cell population, I am still not convinced that this system brings significant improvement compared to the available transduction system, leading to low level of transduction using so high MOI.

We thank the reviewer for their assessment of our work and their suggestions and time. We would like to highlight that due to the absence of intrinsic tropism of VSVdm it is not possible to use functional titration methods and we therefore use an RT-qPCR based quantification of viral RNA genomes in the viral preparation. To reduce the impact of plasmid DNA on the estimations we include a Benzonase treatment step before titration. However, while this approach is applicable to all viral vector preparations, it has been shown to result in overestimates when compared to functional titers (compare <https://www.nature.com/articles/3301731> and <https://doi.org/10.1089/104303403764539387>) as it also measures defective particles. In line with previous estimations, RT-qPCR are on average 200-1000 fold higher than resulting transgene expression. This is also obvious from the Figure 1d, where an MOI of 32 (as determined using the RT-qPCR approach) results in ~30% of transduced cells, which would correspond to a functional MOI of ~0.3. We therefore assume that our calculations of MOIs are ~100-200 fold higher than when using functional titers for MOI calculation.